# Dimension-agnostic and granularity-based spatially variable gene identification using BSP

Juexin Wang [1,2,9] ✉, Jinpu Li [3,4,9], Skyler T. Kramer[3,4], Li Su [3,4], Yuzhou Chang[5,6], Chunhui Xu[3,4], Michael T. Eadon [7], Krzysztof Kiryluk [8], Qin Ma [5,6] ✉ & Dong Xu [2,3,4] ✉

Identifying spatially variable genes (SVGs) is critical in linking molecular cell functions with tissue phenotypes. Spatially resolved transcriptomics captures cellular-level gene expression with corresponding spatial coordinates in two or three dimensions and can be used to infer SVGs effectively. However, current computational methods may not achieve reliable results and often cannot handle three-dimensional spatial transcriptomic data. Here we introduce BSP (big-small patch), a non-parametric model by comparing gene expression pattens at two spatial granularities to identify SVGs from two or three-dimensional spatial transcriptomics data in a fast and robust manner. This method has been extensively tested in simulations, demonstrating superior accuracy, robustness, and high efficiency. BSP is further validated by substantiated biological discoveries in cancer, neural science, rheumatoid arthritis, and kidney studies with various types of spatial transcriptomics technologies.

Spatially resolved transcriptomics (SRT) have been rapidly developed and widely used in biological and biomedical research over the past decade[1–3]. Single-molecule fluorescence in situ hybridization (smFISH) (e.g., MERFISH and SeqFISH + ) and sequencing-based approaches (e.g., 10X Visium)[2] are popular SRT technologies on sliced two-dimensional (2D) samples. A shift has recently occurred towards retaining three-dimensional (3D) positional anatomy at cellular resolution. Wang et al. developed STARmap, which combined an efficient sequencing approach with hydrogel-tissue chemistry for 3D intact tissue RNA sequencing[4], with a throughput of up to 1,000 genes or 10,000 genes[5]. Vickovic et al. developed protocols on consecutive sections to get 3D spatial profiling of rheumatoid arthritis (RA) synovia[6]. Existing works on 3D imaging data construction have significant advantages over 2D data for accurate quantitative interpretation[7]. In contrast to the 2D spatial transcriptomics approach, which depends on sampling strategy (e.g., coronal or sagittal) on sliced samples, the 3D spatial transcriptomics provides a more comprehensive and faithful representation of intact organ structures and functions[8]. It overcomes the inherent 2D bias and enables the visualization of gene expression in relation to the tissue architecture in three dimensions. Such 3D views provide new opportunities in the identification of cell types and states, discovery of new biomarkers, and drug design[9]. However, most current analytic methods are developed and validated on 2D SRT data and cannot be directly applied to diverse types of 3D analyses[10].

[1]Department of BioHealth Informatics, Luddy School of Informatics, Computing, and Engineering, Indiana University Indianapolis, Indianapolis, IN 46202, USA. [2]Department of Electrical Engineering and Computer Science, University of Missouri, Columbia, MO 65211, USA. [3]Institute for Data Science and Informatics, University of Missouri, Columbia, MO 65211, USA. [4]Christopher S. Bond Life Sciences Center, University of Missouri, Columbia, MO 65211, USA. [5]Department of Biomedical Informatics, College of Medicine, The Ohio State University, Columbus, OH 43210, USA. [6]Pelotonia Institute for Immuno-Oncology, The James Comprehensive Cancer Center, The Ohio State University, Columbus, OH 43210, USA. [7]Department of Medicine, Indiana University, Indianapolis, IN 46202, USA. [8]Division of Nephrology, Department of Medicine, Vagelos College of Physicians & Surgeons, Columbia University Irving Medical Center, New York, NY 10027, USA. [9]These authors contributed equally: Juexin Wang, Jinpu Li. ✉e-mail: wangjuex@iu.edu; Qin.Ma@osumc.edu; xudong@missouri.edu

Spatially variable genes (SVGs) are biologically significant as they exhibit variations in expression levels across different regions or cell types within a tissue, indicating their involvement in specific biological processes or functions unique to those regions or cell types[2]. Hence, the inference of SVGs can help researchers gain a deeper understanding of how different cell types and genes contribute to the overall structures and functions of tissues in normal or disease states[11]. Additionally, SVGs can be used as molecular markers to track developmental or disease-related changes in the spatial distribution of specific cell types[12]. Identification of SVGs also facilitates the dissection of biological relationships between spatial organization and molecular cell function, providing critical information for biologists and pathologists. For example, in the mouse olfactory bulb, Stahl et al. discovered functional regions in the mouse brain by identifying SVGs[13], while Maynard et al. discovered laminar and nonlaminar genes in the human dorsolateral prefrontal cortex[14]. The SRT technologies encode the key clues of SVGs, whose expressions rely on the spatial locations of cells[15,16], and identifying SVGs from tens of thousands of genes is often a critical step in analyzing spatial transcriptome data.

Compared to traditional RNA-seq and scRNA-seq analyses that identify differentially expressed genes (DEGs), SVGs in SRT data incorporate gene expression information and corresponding spatial context in geometric coordinates. Similar to DEG identification, SpatialDE[15] and Trendsceek[16] are the first two computational methods for identifying SVGs in 2D SRT studies: SpatialDE utilizes Gaussian process regression to quantify the spatial variance of expression for each gene, while Trendsceek selects SVGs by testing the dependence between the expression and the spatial location for each gene using a permutation process. Afterward, a generalized linear spatial model with Gaussian/periodic kernels, SPARK[17], was proposed to capture the spatial patterns and filter SVGs using the combined $p$-values from each kernel. A simplified version, SPARK-X[18], was later introduced to reduce computational time and memory usage. Recently, nnSVG[19] proposed a scalable approach to identifying SVGs based on nearest-neighbor Gaussian processes. In addition to statistical methods, MERINGUE[10] applied a Voronoi tessellation method to build an adjacency matrix and calculate classical Moran's I score[20] for each gene based on the constructed adjacent matrix to infer SVGs. SpaRTaCo[21] identifies the latent block structure by co-clustering spatial expression profiles of genes. SpaGCN[22] first identifies the spatial domain with graph neural networks and then employs a statistical test to identify SVGs based on the context of inferred spatial domains.

Although some preliminary analysis[4,6] has been conducted on emerging 3D SRT data, significant challenges remain in identifying SVGs in both scales of 2D slices and 3D volume, i.e., the dimension-agnostic SRT data[10,18]. The limited spatial information captured by 2D tissue slices may result in incomplete and biased representations of spatial characteristics, potentially leading to inaccurate biological conclusions[9,23]. Additionally, the existing SVG identification methods require user-defined parameters that can vary across samples and lead to disparate findings that are difficult to justify without prior knowledge of the samples. Hence, a nonparameter method with adequate power is preferred even for 2D data in practical usage. Based on our preliminary analysis, the expression distribution of SVGs tends to exhibit a consistent and specific pattern invariant across different spatial resolutions and views, whereas the expression distribution of non-SVGs has a random pattern with varying characteristics across different views and resolutions. These distributions can be effectively captured by granularity, a concept underexplored in spatial transcriptomics studies. Granularity refers to the extent or hierarchical level to which a material or system comprises distinguishable pieces[24]. We propose that the concept of granularity can be leveraged to identify SVGs in a dimension-agnostic geometric manner. With appropriate quantitative measures, granularity-based criteria can distinguish between biology-informed spatially organized patterns and random patterns.

Here we introduce BSP (big-small patch), a spatial granularity-guided and nonparametric model that enables efficient and robust identification of SVGs from two/three-dimensional SRT data. For each spot in the data, BSP selects a set of neighboring spots within a certain distance to capture the regional means with different granularities. The variances of the expression mean across all spots are then calculated under different scales, and genes with high ratios are identified as the SVGs. One of the unique features of BSP is that it does not make any assumption regarding the distribution of the gene expression levels or the spatial pattern of the spots. The model is robust to fluorescence in situ hybridization (MERFISH, seqFISH +, and STARMap) and sequencing-based (10X Visium and slide-seq) SRT without requiring pre-defined or well-tuned parameters. Compared with existing methods, BSP outperforms other methods for 3D simulations and delivers superior or comparable power and accuracy as current methods for 2D data with a significantly reduced computational cost. In addition, the BSP algorithm is easily implementable, making it versatile and conveniently integrated into various applications, including cancer, neural science, rheumatoid arthritis, and kidney studies. In our experiments on kidney SRT data and 3D RA synovia study, BSP identified several functional-related SVGs with implications to disease mechanisms. In summary, BSP is an accurate, fast, robust, and non-parametric method for identifying SVGs in 2D and 3D SRT data.

## Results

### The big-small-patch algorithm

The proposed BSP algorithm is a granularity-guided approach for identifying SVGs in dimension-agnostic SRT data (Fig. 1). BSP defines a patch for each spot in the SRT data, which includes all neighboring spots within a given radius centered on the spot (Fig. 1A). A pair of patches is then defined, consisting of a small patch with a smaller radius and a large patch with a larger radius (Fig. 1B). This paired big-small patch captures the ambient local expression characteristics in different granularities, delineating spatial patterns in various contexts. Subsequently, the transcriptomic expression variance of the local means is calculated across all pairs of patches, and the ratio between the variance with a big patch and the variance with a small patch is used as the statistic score for each gene. This statistic score can be used to quantify the conservation of SVGs' spatial patterns in different granularities. The distribution of this statistical score is fitted with a lognormal distribution or beta distribution, and genes with statistical significance ($p < 0.05$) in the fitted distribution are defined as SVGs (Fig. 1C).

### BSP can accurately and efficiently identify SVGs in 2D simulations

To demonstrate the effectiveness of the BSP model in analyzing 2D transcriptomic data, we generated a set of simulations following the SPARK[17] framework. We then compared the performances of the BSP model with a basic spatial statistic (Moran's I[20], which is also adopted by MERINGUE[10]) and other established techniques, including SpatialDE[15], SPARK[17], SPARK-X[18], and nnSVG[19]. To ensure a fair comparison of the model performances, we measured their statistical power based on the false discovery rate (FDR), considering the differences between the distribution of calibrated $p$ values from each method. We present three SVG patterns from previous works on the ST mouse olfactory analysis by SpatialDE and SPARK in Fig. 2A. Details of the simulations are outlined in the Methods section.

We evaluated the statistical power of different methods using simulations generated with various signal strengths and noise levels. Signal strengths were measured as the fold-change ($FC$) in cells' expression levels between the pattern and non-pattern areas. To compare the statistical power, we examined different signal-to-noise ratios ($FC = 3,4,5$) with a moderate noise level ($\sigma = 0.5$ as defined in

SPARK) in Supplementary Fig. 1. Additionally, we compared the methods' performance under different noise levels ($\sigma = 0.2, 0.5, 0.8$) with a moderate signal-to-noise ratio ($FC = 4$), as shown in Supplementary Fig. 2. The BSP method consistently showed superior and stable power across a wide range of FDR cutoffs, signal strengths, and noise levels when analyzing the first and third spatial patterns (Fig. 2B). In the second pattern, although the weak and moderate signal strength limits the performance, BSP exhibited better power when the signal strength was high (4-fold) (Supplementary Fig. 2). Compared to other existing approaches, BSP was more powerful, particularly on samples with low signal strengths or high noise levels.

We assessed the computational time and memory usage required for detecting SVGs on 2D data. Compared to existing approaches, BSP performs the SVG analysis with a feasible computational time and memory consumption on personal computers in most scenarios. Computational resource consumption was recorded on an Ubuntu 16.04.4 LTS workstation with Intel(R) Xeon(R) W-2125 CPU @ 4.00 GHz and 32 GB memory in Supplementary Data 1. In analyzing a typical spatial transcriptomic sample with 2,000 spots, BSP was much faster than other existing methods, regardless of the number of genes (Fig. 2C). Similarly, for a spatial transcriptomics sample with 10,000 genes, BSP had the lowest computational time among all methods, despite the number of spots (Fig. 2D). Although not as low as SPARK-X,

the corresponding memory usage was also low, as shown in Supplementary Fig. 3.

## BSP accurately identifies SVGs in 2D space in biological studies

We applied BSP to four previously published 2D spatial transcriptomic datasets, including mouse olfactory bulb[13] and human breast cancer obtained by ST sequencing[13], hippocampus by SeqFISH[25], and mouse hypothalamus preoptic region by MERFISH[26]. We followed the metric evaluation protocols proposed by SPARK and compared Identified SVGs with the provided marker genes in their original research[17]. The results were compared with SpatialDE, SPARK, nnSVG, and Spark-X. All the methods were run with the default parameters.

The mouse olfactory bulb dataset contains 11,274 genes measured on 260 spots using SRT sequencing. BSP detected 9 of 10 marker genes from the original study[13], while SpatialDE detected 3, SPARK detected 8, nnSVG detected 6, and SPARK-X detected 0. Figure 3A and Supplementary Fig. 4 show a comparison between different methods. The only marker gene BSP missed was *Sv2b*, with $p$ values of 0.0518. We reason this missed marker gene has expression variances confined to many isolated, relatively small regions, which could result in the same variances in both big and small patches (Supplementary Fig. 5). The human Breast cancer dataset contains 5,262 genes measured on 250 spots by SRT sequencing. BSP detected 13 of 14 marker genes identified as SVGs

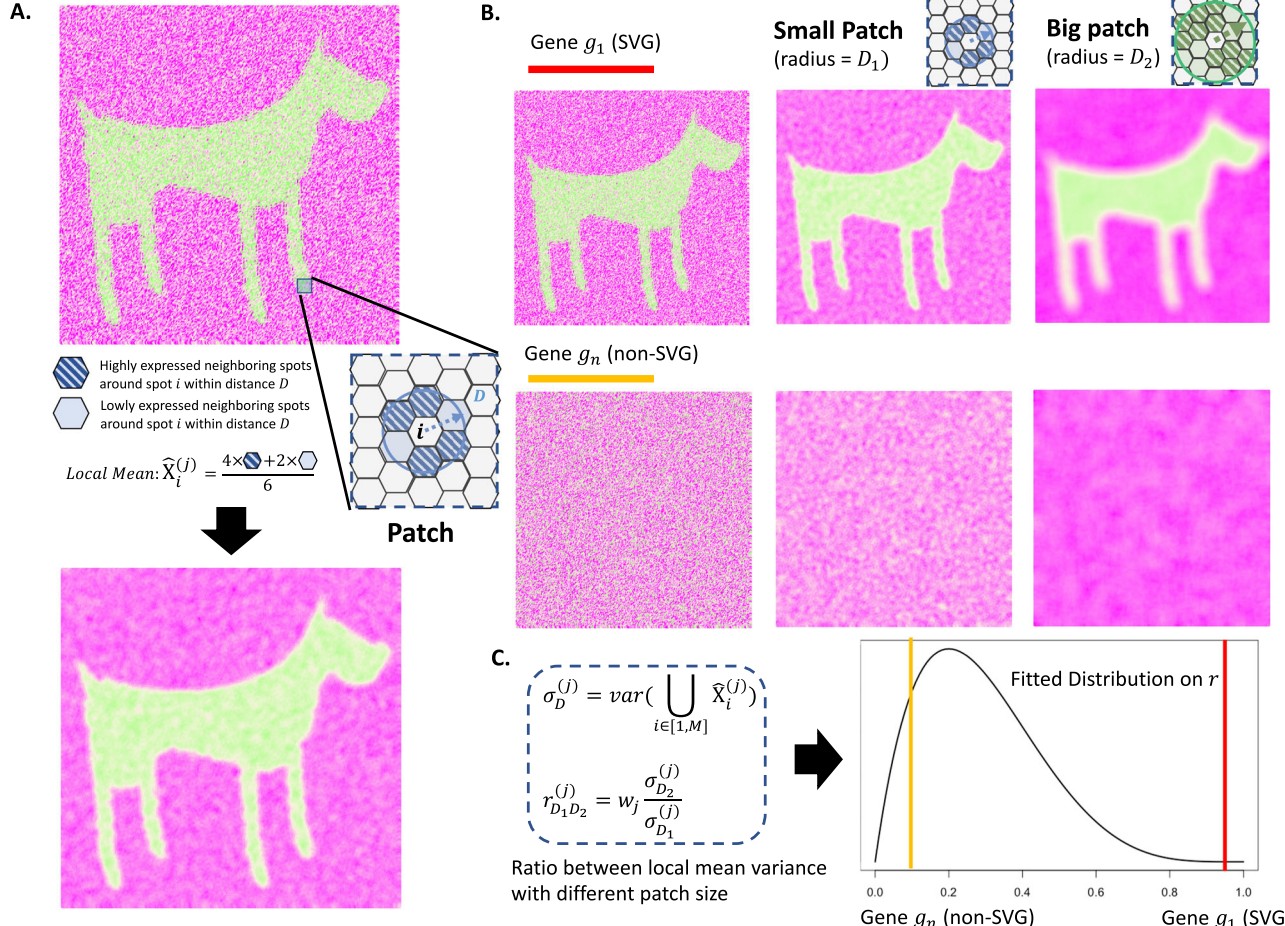

**Fig. 1 | Scheme of BSP. A** Definition of the patch in BSP. For spot $i$, a patch is defined as the set of all its neighboring spots within distance $D$. Local Mean $\hat{X}_i^{(j)}$ is defined as the average expression of gene $j$ for all spots within the patch. **B** Identification of SVGs with BSP. Gene $g_1$ indicates an SVG (red), while $g_n$ is a representative non-SVG gene (orange). Besides the original pattern as the left, we define a small patch of spots with a smaller radius $D_1$ (blue) in the middle column, and a big batch of spots with a larger radius $D_2$ (green) in the right column. **C** The ratio $r_{D_1 D_2}^{(j)}$ of the variance

of the local mean between big batch $\sigma_{D_2}^{(j)}$ and the paired small batch $\sigma_{D_1}^{(j)}$ is chosen as the statistical value $r$. Gene $g_1$ (red bar) is statistically significant compared with the background gene $g_n$ (orange bar) on the fitted distribution of $r$. $w_j$ is the weight to normalize the intrinsic gene expression variance within the gene with the maximum variance, i.e., $w_j = \frac{\sigma_j^2}{\max_k(\sigma_k^2)}$, where $\sigma_j^2$ is the variance of expression levels of gene $j$ of all the spots in the sample, and $1 \le k \le N$. $N$ is the total number of genes. SVG: Spatially Variable Gene.

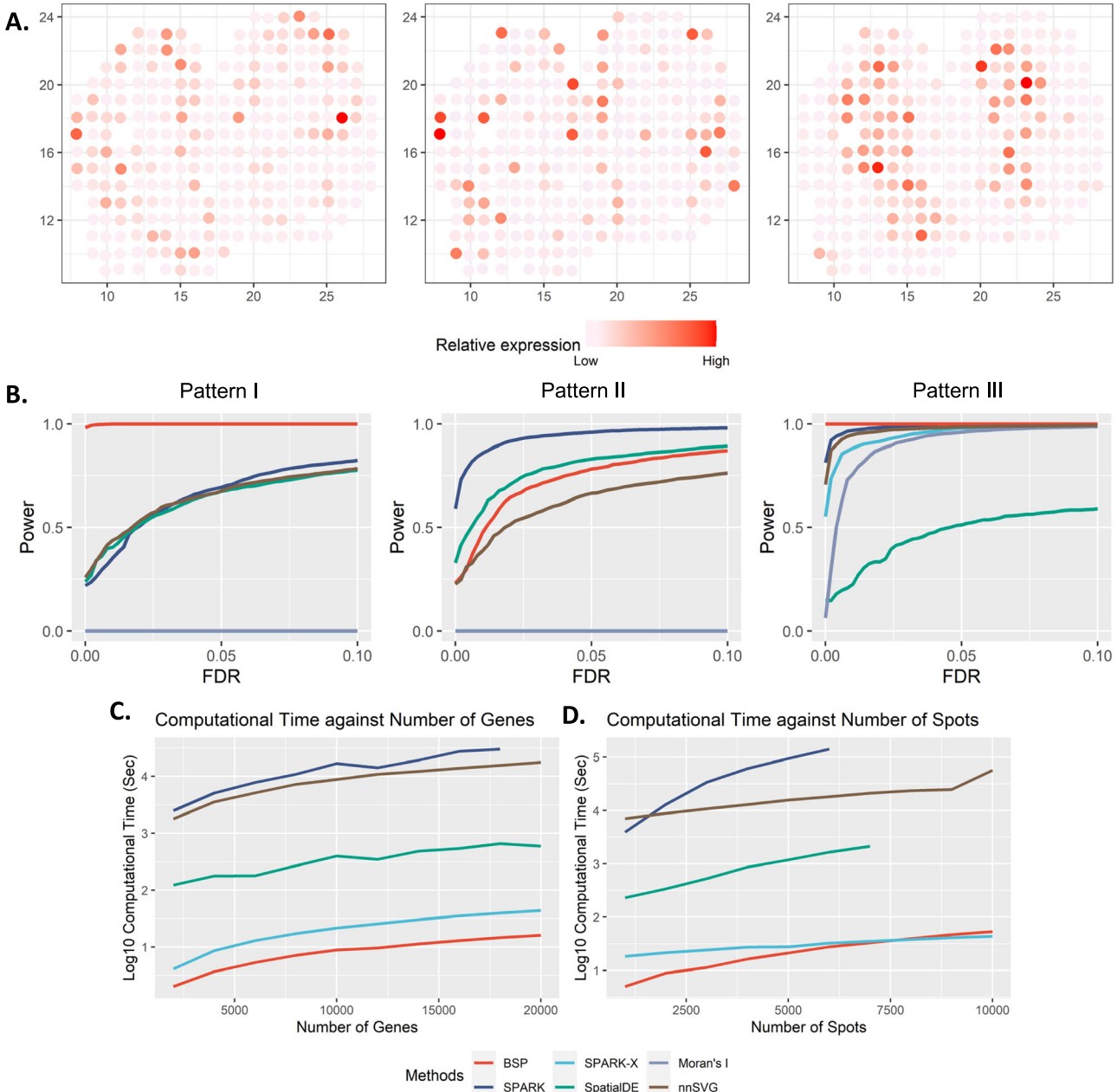

**Fig. 2 | Power analysis for 2D simulation. A** Spatial expression patterns I-III (left to right) as defined in SpatialDE and SPARK. **B** Power comparison among the different methods with moderate noise level ($\sigma$=0.5), and moderate signal strengths (3-folds) for the spatial expression patterns I, II, and III (left to right). All simulations were generated based on the mouse olfactory bulb data with 260 spots of cells. Each simulation replicate contains 1000 SVGs and 9000 non-SVGs. **C** Computational time with an increasing number of genes and fixed 2000 spots. The y-axis indicates the logarithmic of the running time in seconds. **D** Computational time with an increasing number of spots and fixed 10,000 genes. The y-axis indicates the logarithmic of the running time in seconds. The time greater than 48 hours is not shown in the figure. Source data are provided as a Source Data file.

from the original study, while SpatialDE detected 7, SPARK detected 10, and SPARK-X detected 8. The result comparison is shown in Fig. 3B and Supplementary Fig. 6. The marker gene BSP missed was *PIP* with *p* values of 0.0850. The other two FISH-based datasets include the hippocampus dataset, consisting of 249 genes on 131 cells obtained by SeqFISH, and the mouse hypothalamus preoptic region composed of 160 genes on 257 cells by MERFISH. BSP identified most of the marker genes reported in the original studies. Detailed results for mouse olfactory bulb, human breast cancer, hippocampus, and hypothalamus preoptic regions are provided in Supplementary Data 2-5.

We next extended the application of BSP to the study of Acute Kidney Injury (AKI)[27]. We ran BSP on a human kidney biopsy sample with

10X Visium data collected and processed by the Kidney Precision Medicine Project[28]. The biopsy was performed on a 71-year-old Hispanic man two weeks after his initial presentation with severe (stage 3) AKI in the setting of rhabdomyolysis due to a heroin overdose. The biopsy showed acute tubular injury with myoglobin casts (rhabdomyolysis-associated) and diffuse tubular degenerative and regenerative changes, mild interstitial fibrosis, and superimposed C3 mesangial deposits suggestive of resolving infection-related glomerulonephritis.

BSP identified 285 SVGs (*p* value < 0.05) consisting of 317 spots and 14,988 genes. Annotated by clusterProfiler[29], the results were supported by gene ontology (GO) enrichment analysis (Fig. 3C), including relevant enrichments in humoral immune response (*q*-value 1.09e-11), and

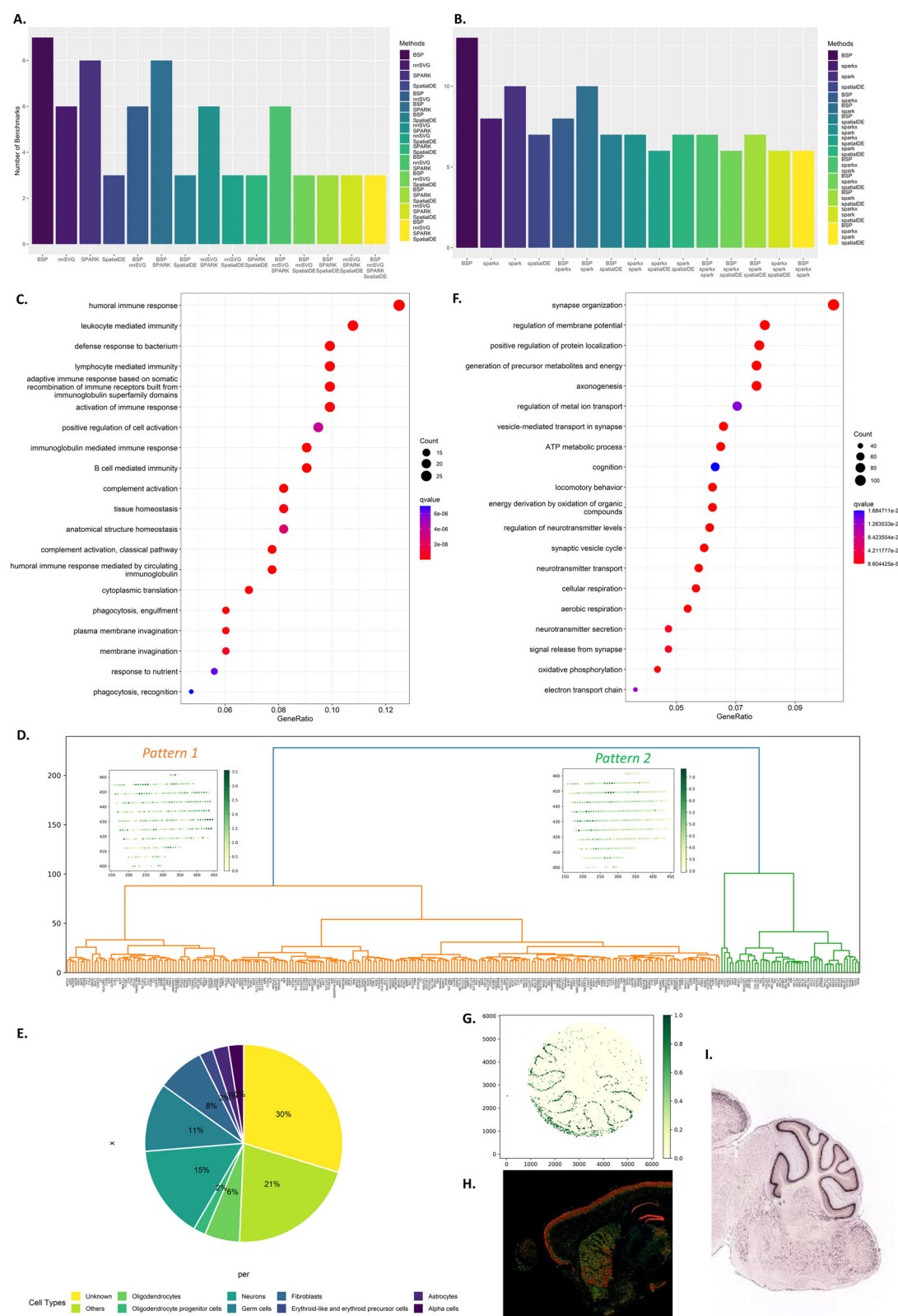

humoral immune response mediated by circulating immunoglobulin (*q*-value 1.63e-10). A Reactome enrichment analysis[30] identified eukaryotic translation elongation (*q*-value 2.77e-13) and influenza infection (*q*-value 5.59e-09), both indicative of the translational shutdown phase of AKI[31] (Supplementary Fig. 7). As innate and adaptive immune responses spatially and temporally correspond with damage to renal tubular cells and recovery from AKI[32,33], these results are consistent with

disease enrichment analysis[34]. Notably, the disease enrichment analysis revealed highly significant terms related to urinary system disease (*q*-value 9.30e-11) and kidney disease (*q*-value 1.36e-10). Supplementary Data 6 lists all the SVG results obtained by BSP. Supplementary Data 7 details the results from GO enrichment analysis, Supplementary Data 8 describes the results from Reactome, and Supplementary Data 9 details the results from Disease Ontology enrichment analysis.

**Fig. 3 | SVGs identified by BSP in biological analysis. A** Number of marker genes identified with different methods in the mouse olfactory bulb study (the original study identified ten marker genes). **B** Number of marker genes identified with different methods in human breast cancer research (the original study identified 14 marker genes). For **A** and **B**, intersections between genes identified from different methods were included in the analyses. **C** Gene enrichment analysis on SVGs identified by BSP on mouse cerebellum data using Slide-seq V2. *q*-values are FDR of gene set enrichment analysis. **D** Hierarchical clustering identified two spatial patterns of SVGs. The left branch (Orange) is Pattern 1, represented by *MT1G* gene in the kidney sample using 10X Visium. The right branch (Green) is Pattern 2, represented by *IGKC* gene in the kidney sample using 10X Visium. **E** Distribution of cell type marker annotations from PanglaoDB on identified SVGs in mouse cerebellum study using Slide-seq V2. **F** Gene ontology enrichment analysis on SVGs identified in acute kidney injury studies. *q*-values are FDR of gene set enrichment analysis. **G** *Calb1* gene expression in the mouse cerebellum data using Slide-seq V2. The expression values were log-transformed, and those greater than 1.0 were normalized to 1.0. **H** Expression and **I** ISH of *Calb1* gene in an adult mouse brain, http://mouse.brain-map.org/experiment/show/75457491. Source data are provided as a Source Data file.

To further investigate the functionalities of SVGs, hierarchical clustering identified two spatial patterns from the kidney sample in Fig. 3D. Pattern 1 includes 235 genes. GO enrichment analysis indicated genes with this expression pattern participated in aerobic respiration (*q*-value 1.04e-09) and oxidative phosphorylation (*q*-value 4.69e-09), consistent with the main pathologic diagnosis of acute tubular necrosis. Pathways of early recovery were also enriched, including kidney development (*q*-value 8.56e-07) and metanephric nephron epithelium development (*q*-value 2.30e-06), which included genes like *PAX8*. Pattern 2 included 50 genes. A GO enrichment analysis indicated genes with this expression pattern were related to humoral immune response (*q*-value 5.15e-05) and tissue homeostasis (*q*-value 5.15e-05). Several immune responses were activated (*q*-value 3.51e-04), including the B cell receptor signaling pathway (*q*-value 4.72e-07). The immune-related pathways are potentially consistent with an inflammatory response to acute tubular injury and potentially resolving infection-related glomerulonephritis. Acute tubular necrosis is characterized by stages of injury, including a transitional stage of translational shutdown, followed by recovery. Both of these pathways were identified in separate localized regions of the kidney biopsy samples and may provide prognostic significance as to the potential for recovery. Thus, our results demonstrate that BSP is able to identify relevant transcripts corresponding to a specific AKI subtype and provide information on the severity (active necrosis and inflammation) and the temporal stage of AKI (evidence of recovery in this case). Supplementary Data 10-13 and Supplementary Figs. 8-11 detail the results from the GO enrichment and KEGG pathway analyses for both patterns. By demonstrating BSP's utility in kidney research, our study highlights the potential for BSP to differentiate the underlying complex disease subtypes in diverse tissue samples.

### BSP identifies SVGs on large-scale spatial transcriptomic studies using feasible computational resources

BSP was tested on three large-scale SRT datasets, including Slide-seq data on mouse cerebellum consisting of 18,671 genes on 25,551 beads, Slide-seq V2 data on mouse cerebellum consisting of 23,096 genes on 39,496 beads, and HDST olfactory bulb data consisting of 19,950 genes measured on 181,367 spots. BSP was successful on these large-scale datasets with a reasonable computational time. On the Ubuntu workstation described in Section 2.2, BSP took 7 and 18 minutes to process the Slide-seq mouse cerebellum and Slide-seq V2 mouse cerebellum data, respectively. The memory costs were around 19GB and 32GB, respectively. For the HDST olfactory bulb data, BSP took 4 hours and 90GB of memory on a High-Performance Computer equipped with Intel Xeon(R) CPU E5-2699 v4 @ 2.20 GHz. The running details are listed in Supplementary Data 14.

BSP detected SVGs with a *p* value less than or equal to 0.05 (*n* = 842, 1156, and 909 in Slide-seq V1 mouse cerebellum data, Slide-seq V2 mouse cerebellum data, and HDST olfactory bulb data, respectively), and we queried PanglaoDB[35] with these detected SVGs. For each of the three implicated datasets, BSP returned numerous neuron-specific and non-neuron-specific genes. Detailed results of detected SVGs from each dataset are listed in Supplementary Data 15-

17. Specifically, in addition to the SVGs corresponding to known cell type composition, many identified genes (30%) were not identified as any cell type markers with PanglaoDB annotations based on the knowledge from previous studies (Fig. 3E).

On 1156 identified SVGs in Slide-seq V2 (Supplementary Data 18), the GO enrichment analysis shows significant enrichments in synapse organization (*q*-value 8.60e-51) (Fig. 3F, Supplementary Data 19). The expression patterns of five representative genes, *Calb1*, *Malat1*, *Nsg1*, *Ttc3*, and *Meg3*, were missed by SPARK-X and annotated as 'unknown' due to the low human brain regional specificity by the Human Protein Atlas[36]. *Calb1* gene (BSP *p* value 2.73e-14, SPARK-X *p* value 0.13, Fig. 3G) is a Ca²⁺ buffering protein found to increase during postnatal development and decrease with aging and neurodegenerative disorders[37]. *Malat1* (BSP *p* value 2.73e-14, SPARK-X *p* value 0.63, Supplementary Fig. 12) is a highly conserved nuclear-retained lncRNA shown to play a role in regulating genes at both the transcriptional and post-transcriptional levels in a context-dependent manner[38]. *Malat1* is shown to be dispensable for normal development and viability in mice[39]. *Ttc3* gene (BSP *p* value 2.73e-14, SPARK-X *p* value 0.30, Supplementary Fig. 13) is known to play a role in cognitive impairment through protein quality control, which is a common phenotype of Down's syndrome and Alzheimer's disease[40]. Another representative gene *Nsg1* (BSP *p* value 2.73e-14, SPARK-X *p* value 1.00, Supplementary Fig. 14), is known to be implicated in regulating endosomal recycling and sorting of several important neuronal receptors[41]. In addition, the *Meg3* gene (BSP *p* value 2.73e-14, SPARK-X *p* value 1.00, Supplementary Fig. 15) modulates AMPA receptor surface expression in primary cortical neurons, and it is in the intricate regulation of the PTEN/PI3K/AKT signaling cascade during synaptic plasticity in neurons[42]. Besides Slide-seq V2, the spatial patterns of *Calb1*, *Ttc3*, *Nsg1*, and *Meg3* were validated by expression (Fig. 3H) and ISH (Fig. 3I) from Allen Brain Atlas[43]. Overall, the structural and functional compartmentalization in the cerebellum revealed by cell type annotation analysis highlights the utility of BSP.

### BSP accurately and robustly identifies SVGs in 3D simulations

We extended the simulation framework in Trendsceek and SPARK further to demonstrate the power of BSP on 3D transcriptomic data. We compared the detection accuracy of SVGs using BSP with that of SPARK-X. The spatial patterns were constructed by a set of center points generated from a random walk with a fixed step length, and any spots within a certain distance from any of the center points were included as the marked cells. We created three continuous 3D patterns, namely, curved stick (Pattern I), thin plate (Pattern II), and irregular lump (Pattern III), controlled by different directions of random walks, as shown in Fig. 4A.

We performed the power analysis based on the FDR, considering the differences in the distribution of calibrated *p*-values. Compared to SPARK-X, BSP demonstrated superior and stable power under a wide range of FDR cutoffs with fixed moderate pattern sizes ($r = 2.0$), moderate signal strength ($FC = 2.5$), and moderate noise level ($\sigma = 1$) for the spatial expression patterns I, II, and III (Fig. 4B). We also varied the pattern sizes, signal strengths, and noise levels while holding the other two parameters constant and found that BSP consistently

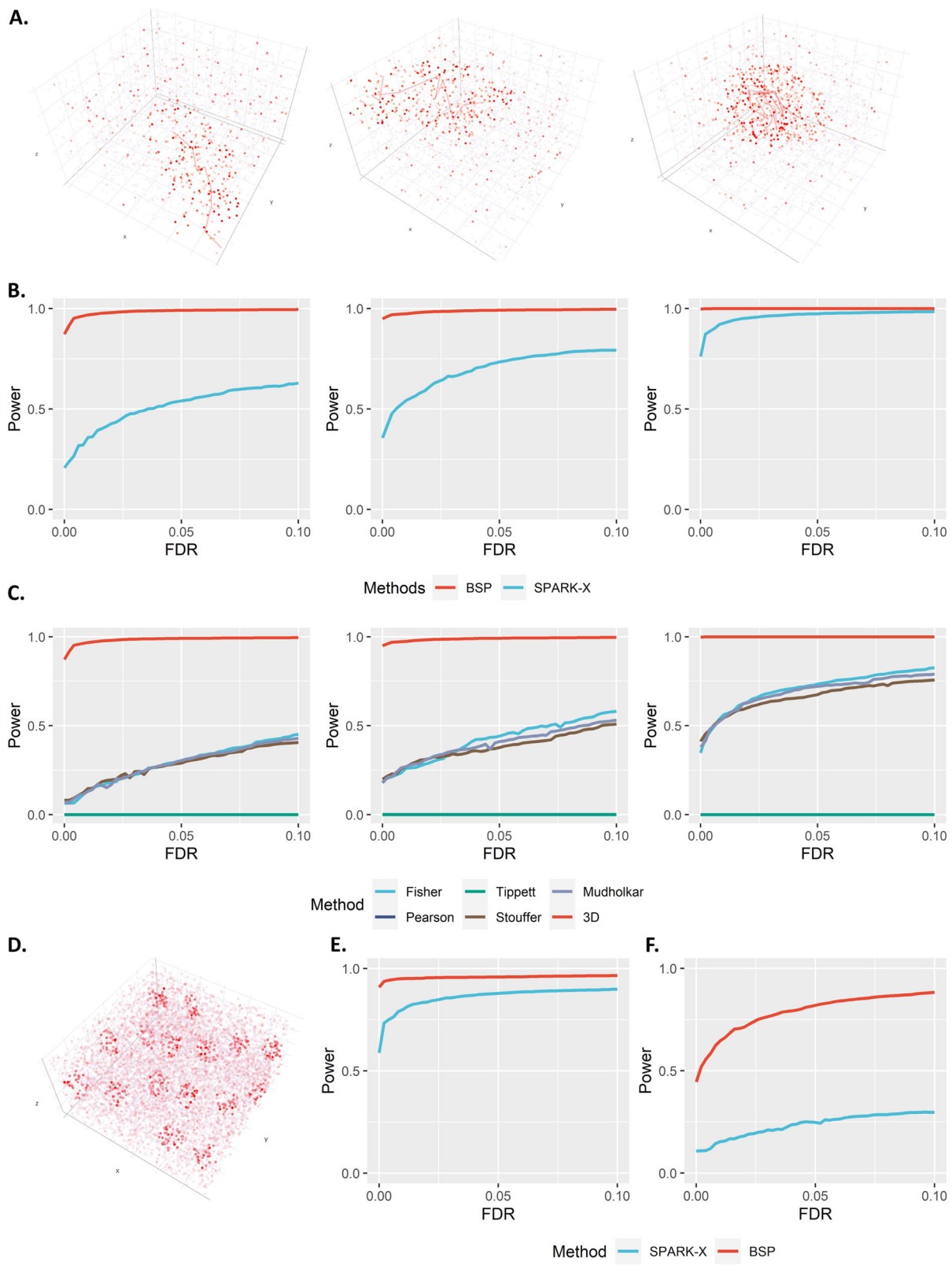

demonstrated greater power in every scenario tested. Supplementary Figure 16 shows the power analysis in different pattern sizes using a fixed moderate signal strength (2.5-fold) and low noise level ($\tau = 0$). Simulations with pattern sizes as small ($r = 1.5$), moderate ($r = 2.0$), and large ($r = 2.5$) are tested on continuous patterns I, II, and III. Supplementary Figure 17 demonstrates the results on different signal strengths using a fixed moderate pattern size (radius of 3) and low noise level ($\tau = 0$). Simulations with signal strengths as low (2-fold), moderate (2.5-fold), and large (3-fold) are tested on continuous patterns I, II, and III. Supplementary Figure 18 shows the results on various noise levels using a fixed moderate pattern size (radius of 3) and moderate signal strength (3-fold). Simulations with high ($\tau = 2$), moderate ($\tau = 1$), and low ($\tau = 0$) noise levels are tested on continuous patterns I, II, and III.

**Fig. 4 | Power analysis for 3d simulation. A** Continuous 3D spatial patterns I-III are controlled by the direction of random walks. **Left**: Pattern I (curved stick): the movements of random walk are monotonic in two directions (x- and z- coordinates, or y- and z-coordinates). **Middle**: Pattern II (thin plate): the movements of the random walk are monotonic in one direction (z-coordinates). **Right**: Pattern III (irregular lump): the movements of the random walk are non-monotonic in any direction. **B** Power comparison of the different methods under varied pattern sizes. Power charts show the averaged true positive rates (y-axis) against the false discovery rates (x-axis) for the detected SVGs using each method. Simulations were performed using fixed moderate pattern sizes ($r = 2.0$), moderate signal strength ($FC = 2.5$), moderate noise level ($\sigma = 1$), and three spatial expression patterns I-III (left to right). All simulations were generated based on the seqFISH data with ten segments (z-coordinate) and 225 spots of cells on each piece (x- and y-coordinates). Each simulation replicate contains 1,000 SVGs and 9,000 non-SVGs.

**C** Power comparisons between 3D SRT and meta-analysis on 2D slices. Simulations were performed using a fixed moderate pattern size, moderate signal strength, and moderate noise level. The red line represents the *p*-value from BSP modeling the 3D SRT. Other colored lines depict *p*-values from various meta-analysis provided by the SciPy package, including Fisher, Pearson, Tippett, Stouffer, and Mudholkar, applied to BSP on 2D slices. Simulations using the 3D pattern I (curved stick), pattern II (thin plate), and pattern III (irregular lump) are shown in the left, middle, and right columns, respectively. **D** Example of simulated discrete spatial patterns in isolated locate domains. **E** Power analysis for identifying SVGs with mixed discrete and continuous patterns. This simulation contains 500 SVGs with discrete patterns, 500 SVGs with continuous patterns, and 9,000 non-SVGs. **F** Power analysis for identifying SVGs in 3D simulations with inconsistent within-plane and inter-plane resolution. FDR: False Discovery Rate. Source data are provided as a Source Data file.

We further investigated the capability of BSP on SRT in the presence of dropout issues[44], where only a subset of the transcriptome is captured by sequencing due to technical limitations. Dropout effects were simulated by randomly assigning zero-values to a certain proportion (10%, 20%, and 30%) of cells in continuous patterns I, II, and III with moderate pattern size, signal strength, and noise level. The power analysis (Supplementary Fig. 19) indicates the robustness of BSP in handling SRT with a reasonable dropout rate.

To demonstrate the advantages of utilizing 3D SRT, we compared SVGs identification performances between direct detection from 3D data and detection through meta-analysis, which combined individual analyses on each 2D slice using various combined probability tests. We performed simulations on all continuous patterns I, II, and III with moderate pattern size, signal strength, and noise level (Fig. 4C). Our findings indicate that employing the BSP model on the 3D data yields superior power than the meta-analysis conducted on the results from 2D slices.

In addition to examining continuous patterns with global influence (depicted by the three 3D patterns in Fig. 4A), we also evaluate the model performances on local discrete spatial patterns (Fig. 4D). These patterns often manifest in isolated, small tissue domains in practical scenarios. Power analysis shows that the BSP model can accurately identify SVGs associated with these local discrete spatial patterns (Fig. 4E). Supplementary Figure 20 represents a comprehensive power analysis for simulations involving varying pattern sizes, signal strength, and noise levels.

In current 3D SRT, especially with data obtained from FISH-based sequencing techniques, it is common for the inter-plane spatial resolution (z-axis) to be considerably lower than the within-plane resolution (x- and y-axes)[4,6]. To further assess the model's performance when the assumption of similar or equivalent spatial resolutions across all three dimensions is violated, we generated 3D simulations with varying scales of spatial resolutions between the within-plane (x- and y-axes) and inter-plane (z-axis) dimensions. Power analysis indicates that the BSP model accurately identifies SVGs in 3D SRT, even in scenarios with discrepancies between the inter-plane spatial and within-plane resolutions (Fig. 4F). Supplementary Figure 21 details a comprehensive power analysis on simulations with varying pattern sizes, signal strength, and noise levels.

Specially, BSP consistently identifies SVGs regardless of selecting different radius values on the scale of the big patch ($D2$) in contrast to the small patch ($D1$). Following co-clustering strategies[21], we set the small patch as the reference unit with a fixed value one, power analysis reveals slight differences when $D2$ ranges from 2.0 to 5.0. The same trends can be observed in all three 3D continuous patterns presented in Supplementary Fig. 22. By making reasonable choices ($D2 \leq 5.0$), BSP exhibits insensitivity to parameter selection, enabling its application across a wide range of SRT platforms and datasets.

Overall, the BSP model demonstrates superior performance in identifying SVGs accurately and robustly in these comprehensive scenarios.

## BSP identifies more meaningful SVGs in the 3D study than stacking results on the 2D analysis

We utilized BSP on two publicly available 3D transcriptomics datasets, mouse visual cortex through STARmap sequencing[4] and human RA synovium using stacking SRT[6]. The STARmap dataset contains 28 known SVGs (23 cell-type markers and 5 activity-regulated genes) measured in 33,598 spots. For these low throughput SRT with few genes, BSP adopted the generated null gene approach proposed by SPARK, and identified all these 28 genes as SVGs.

A study on human RA synovium contains 3D spatial transcriptomic sequencing from six RA patients by stacking 2D slices. Each sample consisted of approximately 13,000 genes on three to seven 2D slices with approximately 1,200 spots in each slice. To evaluate the power of 3D transcriptomics, BSP was first applied to each 2D slice, and then to the stacked 3D volume. Taking the first sample (patient RA1) as an example, we identified 260 genes as the SVGs by intersecting results from four independent analyses on each 2D slice. However, 1,257 genes were detected as the SVGs by analyzing the stacked 3D SRT. All 260 genes from the 2D analysis were included in the gene list detected in 3D space, while 997 additional genes were discovered only in 3D space. We further examined these 997 genes neglected by 2D analysis with the DAVID functional annotations[45] and found significant enrichments in host-virus interaction (Benjamini: *p*-value 4.3e-23), respiratory chain (Benjamini: *p*-value 3.4e-08), innate immunity (Benjamini: *p*-value 2.5e-06), neutrophil degranulation (Benjamini: *p*-value 7.1e-31), and viral process (Benjamini: 7.7e-17) among biological processes.

We also performed a classical meta-analysis by combining four individual analyses on each 2D slice using Fisher's combined probability test with SciPy packages. The meta-analysis identified 804 genes as statistically significant ($p$ value < 0.05). Compared to the 1257 SVGs identified by the 3D analysis, 724 genes were detected as SVGs by both the 2D meta-analysis and 3D settings (Supplementary Data 20), 532 genes were only significant in 3D settings (Supplementary Data 21), and 80 genes were only significant in the 2D meta-analysis setting (Supplementary Data 22). Figure 5A shows the Venn diagram of differences between meta-analysis and 3D analysis. Figure 5B shows GO enrichment analysis on all SVGs identified in 3D settings (Supplementary Data 23). Several immune-related gene ontologies are highlighted in RA studies, including response to interferon-gamma (*q*-value 2.25e-11)[46], myeloid leukocyte migration (*q*-value 3.39e-09), leukocyte migration (*q*-value 3.45e-12), leukocyte chemotaxis (*q*-value 2.29e-09), and regulation of leukocyte migration (*q*-value 5.11e-09)[47]. Supplementary Figure 23 and Supplementary Data 24 show GO enrichment results on 724 genes, both identified by 2D meta-analysis and 3D settings. The same GO enrichment analysis proceeded on 532 genes uniquely identified by 3D settings in Supplementary Fig. 24 and

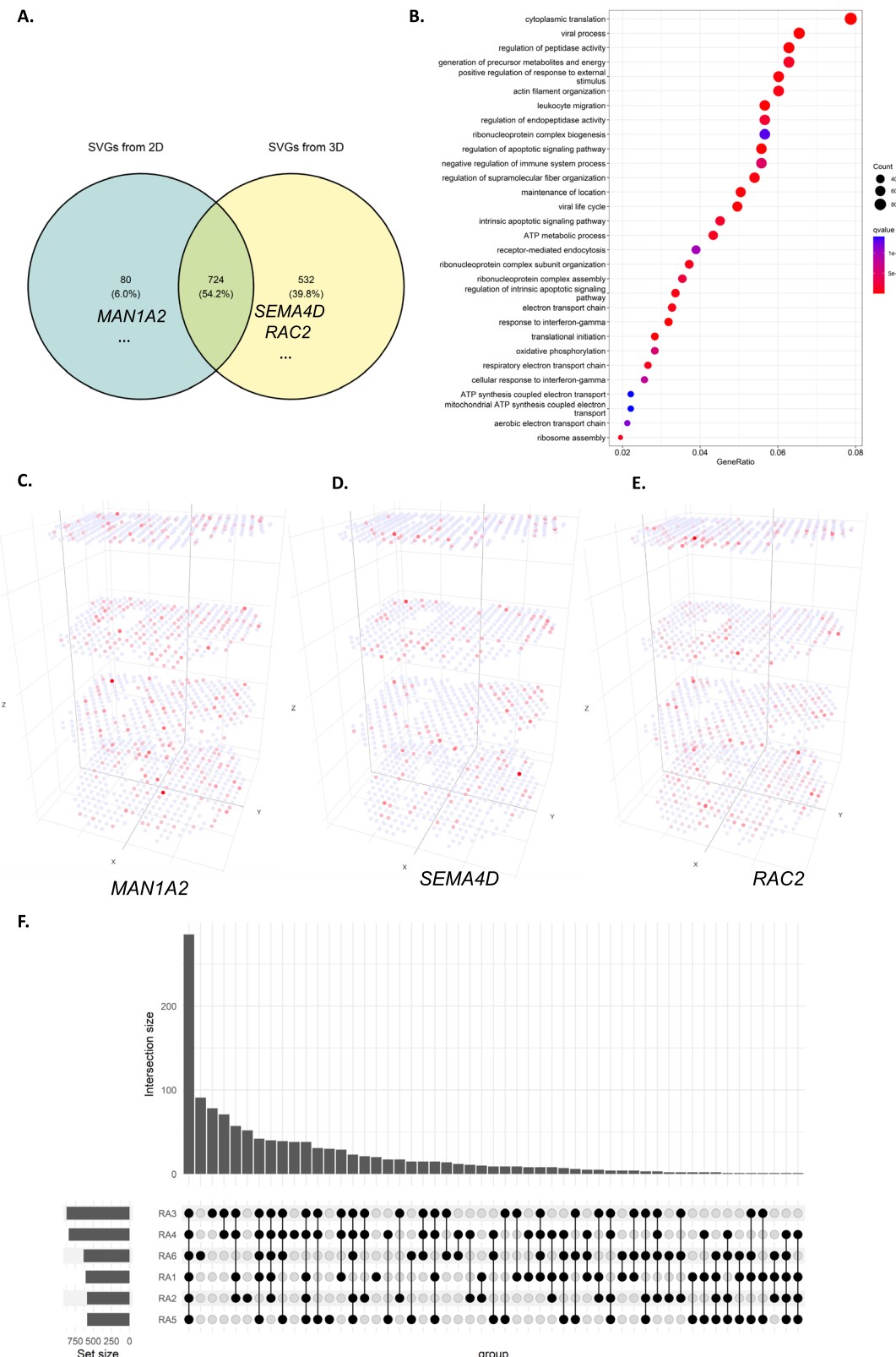

**Fig. 5 | BSP identifies much more meaningful SVGs in the 3D study than stacking results on 2D analysis.** **A** Venn diagram of SVGs identified by 2D meta-analysis and 3D analysis, including representative genes *MAN1A2*, *SEMA4D*, and *RAC2* in **C, D,** and **E**. **B** Gene ontology analysis on SVGs identified in patient RA1 in a 3D setting. **C** *MAN1A2* gene, significant in 2D analysis but not an SVG in 3D setting. **D** *SEMA4D* gene identified as SVGs in 3D transcriptomics but missed by 2D analysis. **E** *RAC2* gene, a very significant SVGs in 3D transcriptomics but missed by 2D analysis. **F** Upset plot of enriched gene ontology terms on all six individual RA patients. SVG: Spatially Variable Gene. RA: Rheumatoid Arthritis. Source data are provided as a Source Data file.

Supplementary Data 25. These highlighted GO terms indicate key immune responses in immunizations in RA progression[48].

2D meta-analysis may lead to some misleading results. Among these genes, *MAN1A2* (Fig. 5C) gets Fisher's combined *p* value 7.56e-08 with four individual 2D *p* values 0.8755, 6.83e-06, 0.0129, and 3.6e-04. However, the *p*-value of *MAN1A2* in 3D settings is 0.3648, making it unlikely as an SVG in 3D space when considering all the slices as a volume. Gene *MAN1A2* is further confirmed by spatial stratified heterogeneity analysis[49] (*p* values 0.2163 for x-coordinates and 0.3189 for y-coordinates), which indicates the absence of statistically significant stratified heterogeneity. On the other hand, among SVGs only significant in 3D analysis, *SEMA4D* plays a role in the immune system, induces B-cells to be aggregated and improves their viability (in vitro)[50]. Although the individual 2D *p* values of *SEMA4D* are 0.9505, 0.9495, 0.9616, and 0.9735, its Fisher's combined *p* value in meta-analysis is 1.0, which has the least possibility of being an SVG in all individuals and meta-analysis on biased 2D analysis. However, the BSP test results of gene *SEMA4D* is 0.0482 on the 3D volume, making it stand out from the genes (Fig. 5D). A spatial stratified heterogeneity analysis further confirmed *SEMA4D* is statistically significant stratified heterogeneous (*p* values 0.0019 for x-coordinates and 0.0004 for y-coordinates). Another example that fails in the 2D analysis is *RAC2* (Fig. 5E), which encodes a member of the Ras superfamily of small guanosine triphosphate (GTP)-metabolizing proteins involved in generating reactive oxygen species. Although Fisher's combined *p* value is 0.0942 with four individuals as 0.1173, 0.0846, 0.3286, and 0.3500, it is very significant with a *p* value of 4.4929e-15 in the 3D setting.

To explore the functionalities of SVGs further, we performed hierarchical clustering on the RA1 sample, resulting in the identification of four distinguishable spatial patterns (Supplementary Fig. 25). Pattern 1 consisted of 91 SVGs and exhibited enrichment in ribosome genes associated with biogenesis pathways (*q*-value 3.10e-16). This pattern suggests a general biosynthetic functionality supporting RA tissue across the highlighted regions[51]. Pattern 2 comprised 774 SVGs, showing enrichment in lymphocyte proliferation (*q*-value 2.22e-07) and activation of the immune response (*q*-value 1.69e-06), indicating an immune activation region. Notably, the presence of *CXCL13* and *MS4A1* in this pattern suggests the onset of inflammation in RA[52]. Pattern 3 consisted of 120 SVGs and enrichment in apoptosis activity (*q*-value 7.31e-06) and complement activation process in the pathogenesis of RA (*q*-value 1.96e-05), supported by genes such as *C1S*, *C1R*, *C1QC*, *S100A8*, and *S100A9*[53]. Similarly, Pattern 4 included 233 SVGs and showed enrichment in apoptosis (*q*-value 4.33e-05) and regulation of leukocyte migration (*q*-value 1.11e-03), indicating the formation of chronic inflammation and autoimmunity in RA development through recruiting leukocytes[54]. Supplementary Data 26-33 and Supplementary Figs. 26-33 provide detailed results from the GO enrichment and KEGG pathway analyses for all four patterns. This analysis underscores the opportunities afforded by BSP analysis on intact 3D volumes in identifying SVGs compared to potentially biased 2D analysis.

The same analyses were conducted on each of the six RA patients individually. The enriched GO terms of 3D SVGs identified in each patient were presented in Fig. 5B and Supplementary Figs. 34-38. We observed that most GO terms were consistently enriched in all the patients (Fig. 5F), indicating that BSP robustly identified 3D SVGs across various samples.

## Discussion

Advances in spatial transcriptomics have facilitated the measurement of high-throughput multi-cellular- or cellular-level gene expression in the spatial context. This fast-growing 3D technology is critical for understanding the relationship between tissue structure and underlying biological function, posing new challenges in identifying SVGs vital in linking individual genes to spatial expression variance. The proposed BSP provides a dimension-agnostic and utilizes a big-small

patch algorithm to identify SVGs at varying levels of granularity. The performance of BSP has been validated in both simulations and real studies using 2D and 3D data. While there is still a debate over the gold standard for the definition of SVGs in biological studies, we follow the protocol adopted by SPARK for power analysis and biological annotation. Notably, simulations provide an alternative benchmark for methods development. In the 2D simulation, BSP outperformed existing methods in most scenarios with different signal-to-noise ratios. In the 3D simulations, BSP demonstrated its superiority compared to other well-known criteria, such as Moran's-I. Meanwhile, these 3D simulations can serve as benchmarks for developing new methods. In biological studies using 2D and 3D data, BSP identified more convincing SVGs than existing methods with good control of false positives. For instance, in a human RA study, BSP revealed that analyzing SVGs as a volume in 3D data outperformed stacking results on individual 2D slices.

The innovation of BSP lies in its dimension-agnostic and granularity-guided approach, which utilizes paired big-small patches. Intuitively, the big patch provides a global view of the spatial pattern with a lower resolution, while the small patch focuses on the local details with a higher resolution. Using ratios between variances of the paired patches, BSP can accurately delineate the spatial patterns in a quantitative manner. Specifically, BSP operates by assessing how rapidly the variances of local means change as the radius of the patch is adjusted. The primary source of these variance fluctuations arises from neighboring regions that exhibit distinct expression levels, where the local means within such regions change more gradually as the patch radius varies than between neighboring cells or spots. Consequently, the velocity of changes in the variances of local means for a gene with global spatial patterns is comparatively slower than for the genes that lack discernible spatial patterns. This behavior becomes notably prominent when the patch radius is chosen within a reasonable range. Although the patch radius is significantly smaller than the spatial pattern, the patches are averaged over all positions on the whole spatial transcriptomic space and hence the model can capture the global patterns of SVGs. Therefore, we recommend using the default value of 3.0 for the radius of large patches in most situations, especially when the sizes of spatial patterns remain unclear. Users may adjust the value of D2 within the range of 2.0 to 5.0, particularly when dealing with a known, large-scale spatial pattern. Moreover, this approach is applicable to any dimension. These defined patches can effectively capture the characteristics of the expression patterns in both 2D space and 3D volume, making BSP capable of analyzing SRT data in both dimensions.

This granularity-guided approach makes BSP a data-driven and non-parametric model. First, BSP is particularly well-suited for the complexities of biological data, especially in the tumor microenvironment, where fixed spatial patterns cannot be assumed to form locally and globally. BSP's effectiveness in these complex scenarios has been demonstrated in both 2D and 3D simulations, without preconceived assumptions about the underlying distributions. Second, BSP is robust to different levels of signal strengths and tolerates occasional noise. It robustly discovers the same persistent results in different samples, as the spatial patterns are invariant in different scales. Third, the BSP algorithm is highly efficient. In the typical scenario of a 10X Visium scale, BSP remains the fastest method among all the existing methods. Even for large-scale datasets, such as Slide-seq, Slide-seqV2, and HDST, BSP remains feasible with reasonable computational resources. Fourth, BSP's core implementation is just a few dozen lines of code, making it easy to implement and adaptable to different usage scenarios.

Although BSP has shown notable advancements in quantitatively measuring spatial patterns using the lognormal distribution to fit the distribution of test scores of all the genes, some limitations still need to be addressed. Through meticulous examination of histograms on permuted data vs. density of distribution and corresponding Q-Q plots

(Supplementary Figs. 39-40), and goodness-of-fit test with Cramer-von Mises criterion (Supplementary Data 34-35), it becomes evident that the lognormal distribution offers a more suitable fit than the beta distribution for studies involving 2D simulation data, mouse olfactory data, and human breast cancer data. However, alternative statistical distributions or non-statistical ranking measurements could be explored to further improve the fitting of the distribution of ratios between variances of the averaged expression in the paired big-small patch. In the practical usage where spatial patterns exhibit alternating high- and low-expressed cells within a confined area, e.g., a pattern of thin curved stick in Fig. 4A, it is essential to exercise caution regarding the choice of patch radius. Using an excessively large patch radius can potentially result in a reduction in statistical power, as demonstrated in Supplementary Fig. 22. Furthermore, BSP compromises the performance and computational resources in SRT studies. Although it performs better on the benchmarks, BSP consumes more time and memory than SPARK-X on large-scale datasets.

In conclusion, BSP has demonstrated its efficacy as a robust method for identifying SVGs in both 2D and 3D spatial transcriptomics analysis. As 3D sequencing technologies continue to advance and mature, we anticipate BSP to be increasingly valuable in future applications of 3D spatial transcriptomics. We will also explore the incorporation of sparse matrices to accelerate the computational processes on large-scale data, especially high-resolution spatial transcriptomics data. Moreover, as time is often considered as the fourth dimension in development biology[55,56], we will also explore the potential for spatiotemporal studies using BSP.

## Methods
### BSP algorithm
BSP aims to identify spatially variable genes in 2D or 3D SRT data. The algorithm contains several steps, including (1) normalizing expression and spatial coordinates, (2) defining big and small patches for each spot based on neighboring spots with a larger or small radius, (3) calculating local means of gene expression for both big and small patches, (4) computing the ratio between the variances of local means between big and small patches for each gene, and (5) fitting the ratio of each gene with a log-normal distribution and calculating the $p$-value for each gene. The flowchart of the BSP algorithm is shown in Supplementary Fig. 41.

### Problem setting and data normalization
On an SRT sample with $M$ spots and $N$ genes. The coordinates of spot $i$ are $(x_i, y_i)$ for 2D spatial transcriptomics, $(x_i, y_i, z_i)$ for 3D spatial transcriptomics. The expression level of gene $j$ in spot $i$ is denoted as $X_i^{(j)}$, where $1 \leq i \leq M$, $1 \leq j \leq N$. The goal of BSP is to identify SVGs from all $N$ genes with significant spatial patterns.

All gene expression levels are normalized and scaled to [0,1] using a min-max normalization across all spots, $0 \leq X_i^{(j)} \leq 1$ for all $X_i^{(j)}$. The normalization of spatial coordinates of the spots on SRT is based on the density of spots. The coordinates of spots in each direction are divided by the estimated density, which is calculated as the total number of spots divided by the area (2D) or volume (3D) of the sample. For simplicity, a rectangle is defined as the 2D space, and a cube is defined as the 3D volume. The rescaling functions $f$ for 2D space is defined as Eqs. (1), and for 3D space as Eq. (2):

$$f(x_i, y_i) = \sqrt{\frac{N}{\Delta X \Delta Y}} \cdot (x_i, y_i) \qquad (1)$$

$$f(x_i, y_i, z_i) = \sqrt[3]{\frac{N}{\Delta X \Delta Y \Delta Z}} \cdot (x_i, y_i, z_i) \qquad (2)$$

where $x_i$, $y_i$, and $z_i$ are the coordinates of spot $i$. $\Delta X$, $\Delta Y$, and $\Delta Z$ denote the ranges of the sample space. They can be calculated as the

differences between the maximum and minimum coordinates in each direction for the cube, as: $\Delta X = \max(x) - \min(x)$, $\Delta Y = \max(y) - \min(y)$, and $\Delta Z = \max(z) - \min(z)$.

This spatial coordinate normalization step ensures an adequate number of spots captured by the pre-defined radii $D_1$ and $D_2$. The goal of this step is to minimize the average spot-to-spot distance to slightly less than one unit. Typically, the default value of $D_1$ is set as one unit to capture the nearest neighbors, while $D_2$ is set to three units to include more spots in the patches. More comprehensive exploration shows the BSP performance is insensitive in selecting $D_2$ on scenarios of simulations (Supplementary Fig. 22) and on real data of mouse olfactory bulb and human breast cancer studies (Supplementary Data 36).

### Big-small path
After coordinates normalization, the Euclidean distance between spots $i_1$ and $i_2$ is calculated as $dist(i_1, i_2)$. For a given spot $i$, a **patch $S_i$** is defined as the set of neighboring spots $l$ within the radius of $D$

$$S_i = \{l : dist(i, l) < D, l = 1 \leq l \leq M \text{ and } l \neq i\}$$

With a patch $S_i$, $\hat{X}_i^{(j)}$ is defined as the **Local Mean**, the average expression level of gene $j$ in this patch.

$$\hat{X}_i^{(j)} = \frac{\sum_{i \in S_i} X_i^{(j)}}{num(S_i)} \qquad (3)$$

where $num(S_i)$ is the cardinal number of spots within $S_i$. Local Mean describes the expression characteristics in the patch. $\sigma_D^{(j)}$ is defined as the variance of Local Means of all patches on gene $j$.

$$\sigma_D^{(j)} = var\left(\bigcup_{i \in [1, M]} \hat{X}_i^{(j)}\right) \qquad (4)$$

For a gene $j$ without any spatial expression pattern, i.e., the distributions of the expression levels being identical across all spots, $\sigma_D^{(j)}$ equals to 0. Otherwise, $\sigma_D^{(j)} > 0$. If distance $D$ is big enough to cover the radius of the sample, then $\sigma_D^{(j)}$ also equals 0 for each patch containing all spots, as the Local Means are the same for each spot.

For each spot $i$, we define a paired big-small patch, i.e., a mall patch is defined as $S_i'$ with a radius $D_1$, a big patch is defined as $S_i''$ with a radius $D_2$, where $D_1 < D_2$. We take the $r_{D_1 D_2}^{(j)}$, the ratio between the variances of the paired local averaged expression levels between big patch and small patch, describes the characteristics of the spatial pattern on gene $j$, defined as:

$$r_{D_1 D_2}^{(j)} = w_j \frac{\sigma_{D_2}^{(j)}}{\sigma_{D_1}^{(j)}} \qquad (5)$$

$w_j$ is the weight to normalize the intrinsic gene expression variance within the gene with the maximum variance, i.e., $w_j = \frac{\sigma_j^2}{\max_k(\sigma_k^2)}$, where $\sigma_j^2$ is the variance of expression levels of gene $j$ of all the spots in the sample, and $1 \leq k \leq N$.

### Fitting distribution and calculating $p$ value
After $r_{D_1 D_2}^{(j)}$ is calculated with all the genes, $j \in [1, N]$, a lognormal distribution is approximated for the distribution of $r_{D_1 D_2}^{(j)}$ using the *stat* packages from sklearn[57]. Depending on the characteristics of the data, a beta distribution is considered as an alternative approximation for $r_{D_1 D_2}^{(j)}$ using the same packages. The null hypothesis that a gene has no spatial pattern is thus reformulated as the ratio of a gene adhering to the fitted log-normal or beta distribution. To tolerate the potential noise and long-tail deviations, a one-sided $p$-value is assigned to each gene if $r_{D_1 D_2}^{(j)}$

exceeds the upper tail of the fitted distribution at a probability of $100*(1 - \alpha)\%$, where $\alpha$ refers to the significance level (usually set as 0.05). We assume that only a small portion of genes located in the tail of the distribution are SVGs, while the majority of genes are non-SVGs that are spatially independent. This hypothesis is particularly applicable in high-throughput SRT platforms like 10X Visium, which involve more than thousands of genes. However, in low-throughput SRT platforms such as FISH, where there are insufficient genes (hundreds or even less), a set of random genes permuted across spatial locations as null genes are generated to compensate for non-SVGs. These null genes help estimate the complete distribution for practical usage.

## 2D simulation on mouse olfactory bulb data

We utilized the mouse olfactory bulb data within the framework of SPARK to construct 2D simulations. The simulation was based on mouse olfactory bulb data consisting of three spatial expression patterns measured on 260 spots (as shown in Fig. 2A). Each simulation contained 1000 simulated SVGs with identified patterns in SpatialDE and SPARK, as well as 9000 non-SVGs generated through gene permutation without any spatial expression pattern. The $p$ values from basic spatial autocorrelation statistics Moran's I, SpatialDE, SPARK, SPARK-X, and BSP were calculated to quantify the corresponding power (true positive rates) given a false discovery rate (FDR). To illustrate the rate of true positives (y-axis) identified by each method at different FDRs (x-axis) in power analysis, we generated ten replicates for each simulation. Specifically, simulation data was generated under different signal-noise ratios ($FC = 3,4,5$) with a medium level of noise ($\tau = 0.5$, as defined in SPARK). Then another set of simulation data was generated under different noise levels ($\tau = 0.2, 0.5, 0.8$) with a moderate signal-noise ratio ($FC = 4$).

## 3D simulation on FISH data

**3D simulation on FISH data with continuous spatial patterns.** For the simulations in 3D space scenarios, we extended the framework originally introduced by Trendsceek and SPARK. All simulations were generated based on seqFISH data, with 10 segments in the z-coordinate and 225 spots representing cells in each piece in the x- and y-coordinates. We assume the sample was cryosectioned into 10 sections, with each section placed on an individual array without any direct contact between array surfaces. To generate spatial locations for a fixed number of cells ($n = 225$) in each section, we used a random-point-pattern Poisson process. These spatial locations for each section were then stacked together with the index of the section serving as the z-coordinates ($z = 1, 2, \ldots, 10$).

The 3D spatial patterns were constructed using a set of spheres with center points generated through a random walk with a fixed step length of 2. We included three types of continuous spatial patterns in the simulations by controlling the range of directions (Fig. 4A). These continuous patterns include Pattern I (curved stick), the movements of a random walk are monotonic in two directions (x- and z- coordinates, or y- and z- coordinates); Pattern II (thin plate), the movements of a random walk are monotonic in one direction (z- coordinates); Pattern III (irregular lump), the movements of a random walk are non-monotonic in any directions. We produced 1000 SVGs with 3D patterns for each simulation and generated 9000 non-SVGs without any spatial expression pattern by permuting known patterns.

The expression of SVGs was sampled based on whether the cell was inside or outside the pattern, distinguishing between marked and non-marked cells. For marked cells inside the pattern, we randomly selected gene expression values from the upper quantile of the gene expression distribution in the seqFISH data. For non-marked cells and those outside the pattern, we assigned gene expression randomly from the expression measurements in the seqFISH data. Non-SVGs were generated by permuting gene expressions of SVGs. For each SVG, the expression values were permuted and repeated 9 times (i.e., randomly

assigning values to all cells without replacement). Finally, random noise was added proportionally to the averaged standard deviation of expressions in all genes.

To systematically explore the influences under different scenarios, we held two parameters constant while manipulating the third to vary the patterns' sizes, signal strengths, and noise levels. We tested three sphere radius values ($r$) of 1.5, 2.0, and 2.5, which determined the pattern size. Quantile thresholds of 0.66, 0.80, and 0.88 were set, corresponding to expected expression fold changes ($FC$) of 2.0, 2.5, and 3.0 between marked and nonmarked cells, indicating low, moderate, and strong signal strengths, respectively. We applied random noise following a Gaussian distribution with mean zero and the standard deviation ($\tau$) of 0, 1, and 2 times the averaged standard deviation of the expressions of all simulated genes to represent low, moderate, and high noise levels. In the 3D simulations, we varied the pattern sizes ($r = 1.5, 2.0, 2.5$), expression fold changes ($FC = 2.0, 2.5, 3.0$), and noise levels ($\tau = 0, 1, 2$) across continuous spatial patterns I, II, and III. For each combination of pattern size, signal strength, and noise level, we conducted 10 replicates to perform the power analysis.

**3D simulation on FISH data with discrete spatial patterns.** To further evaluate the model's performance on data with locally influential discrete patterns (Fig. 4D), we designed a set of 3D simulation scenarios within the ranges of x- and y- coordinates from 0 to 30. The local discrete spatial patterns were constructed using solid spheres with a center-to-center distance of 8 units. Specifically, sixteen center points were selected, with fixed z-coordinates of 5.5, and the x- and y-coordinates were generated from a sequence of numbers from 3 to 27 with an interval of 8. To introduce randomness into spatial patterns, we incorporated a uniformly distributed random variable ranging from −2 to 2 and added it to the coordinates of each center point. The cells within the spheres were marked, and expression values were assigned as Section 4.3.1.

In these simulations, SVGs were generated using the Irregular lump pattern described in Section 4.3.1. We created 500 SVGs with locally influential discrete patterns and an additional 500 SVGs with globally influential continuous patterns, along with 9000 non-SVGs as permutated genes without any spatial pattern. We considered three scenarios to compare the effects of pattern sizes, signal strengths, and noise levels, respectively. We adjusted radius values from 1.5 to 2.0 (small and moderate), expected expression fold changes from 2 to 2.5 (small and moderate), and noise levels from 2.0 to 3.0 (moderate and high). To ensure robustness, ten replicates were generated to perform the power analysis.

**3D simulation on FISH data with varying spatial resolutions in the z-axis.** In practical scenarios, the assumption of similar or equivalent spatial resolutions across all three dimensions may not hold, particularly in FISH-based sequencing techniques where the inter-plane spatial resolution (z-axis) is often significantly lower than the within-plane resolution (x- and y-axes). To systematically test the capability of BSP on 3D spaces, we increased the resolution of the z-axis by multiplying the z-coordinates by ten ($z = 10, 20, \ldots, 100$) and replicated the simulation as described in Section 4.3.1. Considering the substantial differences between the z-coordinate and x- or y-coordinates, we fixed the direction of random walks in the z-axis to ensure that the sliced planes captured the simulated spatial pattern (Supplementary Figure 42). We varied the pattern sizes (small and moderate radius = 1.5 or 2.0), signal strengths (small and moderate, fold change = 2.0 or 2.5), and noise levels (small and moderate, $\tau = 1$ or 2), respectively. The performance of the BSP model was compared with SPARK-X.

## Biological data collection and analysis

For studies on mouse olfactory bulb, human breast cancer obtained by SRT sequencing, hippocampus by SeqFish, and mouse hypothalamus

preoptic region by MERFISH, we followed the analysis protocol adopted by SPARK. For studies on Slide-seq data, Slide-seqV2 data, and HDST data, we followed the analysis protocol adopted by SPARK-X.

In the case of human RA synovium studies, the spatial locations in 2D slices were normalized with unit one. These 2D slices were stacked together with interval one on the z-axis to construct a volume on 3D transcriptomics. Analysis was performed based on the normalized data provided by the authors.

For kidney analysis, all the data were generated using 10X Visium platforms and processed with CellRanger. Expression data is quality-controlled and preprocessed by Seurat with scTransform[58]. Hierarchical clustering is performed by SciPy from the sklearn package[57] (Version 1.1.2) in Python 3.9.12.

### Annotations

The annotations using PanglaoDB were performed by rPanglaoDB (Version 0.2.1). Go enrichment analysis was performed by topGO[59] (Version 3.16). The Reactome pathway analysis was performed by ReactomePA[30] (Version 3.16). Disease Ontology Semantic and Enrichment analysis was performed by DOSE[34] (Version 3.16). The meta-analysis was performed by SciPy from the sklearn package[57] (Version 1.1.2) in Python 3.9.12.

### Statistics & reproducibility

In study design, no statistical method was used to predetermine sample size. No data were excluded from the analyses. The experiments were not randomized, and the Investigators were not blinded to allocation during experiments and outcome assessment.

### Reporting summary

Further information on research design is available in the Nature Portfolio Reporting Summary linked to this article.

## Data availability

All relevant data supporting the key findings of this study are available within the article and its Supplementary Information files. The mouse olfactory bulb and human breast cancer data are available at (https://www.spatialresearch.org/resources-published-datasets/doi-10-1126science-aaf2403/), the MERFISH data can be downloaded from https://datadryad.org/stash/dataset/doi:10.5061/dryad.8t8s248, and the SeqFISH data is available at https://www.cell.com/cms/10.1016/j.neuron.2016.10.001/attachment/759be4dc-04a6-4a58-b6f6-9b52be2802db/mmc6.xlsx. Slide-seq data, Slide-seqV2 data, HDST data, and human rheumatoid arthritis synovium data are available at Broad Institute's single-cell repository with ID SCP354, SCP948, SCP420, and SCP1414. The STARmap data set is available at https://github.com/drieslab/spatial-datasets/tree/master/data/2018_starmap_3D_cortex. The kidney spatial transcriptomics data can be downloaded from the Kidney Tissue Atlas (https://atlas.kpmp.org/) with ID 32-10074 [https://doi.org/10.48698/3z31-8924]. The generated simulation data has been deposited in Figshare database at https://doi.org/10.6084/m9.figshare.24187923[60]. Source data are provided with this paper.

## Code availability

The source code of BSP is freely available at https://github.com/juexinwang/BSP/[61].

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

## Acknowledgements

This work is supported by National Institutes of Health grants R01DK138504 (to JW, QM, and MTE), R35GM126985 (to DX), R21HG012482 and U54AG075931 (to QM), U54DK134301 and U01DK114923 (to MTE), 2R01DK105124, 1R01LM013061, U01HG013201 (to KK), the AnalytiXIN initiative (to JW), as well as the Pelotonia Institute of Immuno-Oncology (PIIO) (to QM).

## Author contributions

Conceptualization: J.W., J.L., Q.M., and D.X.; methodology: J.W., J.L., and D.X.; software coding: J.W. and J.L.; data collection and investigation: S.K., L.S., Y.C., and C.X.; data analysis: J.L, J.W., S.K., L.S., and Y.C.; pathology analysis: M.E, K.K, and Y.C.; software testing and tutorial: J.W., J.L., and L.S.; manuscript writing, review, and editing: J.W., J.L., Q.M., and D.X.

## Competing interests

The authors declare no competing interests.
