## [Peer Review File · Nature Communications]

Dimension-agnostic and granularity-based spatially variable gene identification using BSPReviewer #1 (Remarks to the Author):

See all the comments in the attached file

Reviewer #1 Attachment on the following page

Review of “Dimension-agnostic and granularity-based spatially variable gene identification”

The article introduces BSP, a model for detecting spatially variable genes (SVGs) using spatial transcriptomic experiments. The idea is to propose a fast and flexible way to determine SVGs in 2D spatial transcriptomic protocols (e.g., MERFISH, SeqFISH+, 10X Visium) and to fill the gap of missing methods for determining SVGs in 3D protocols, which are becoming more common.

I find the idea of using a granularity-based method interesting and promising. I see the BSP model as a non-parametric model that is free from any particular modeling assumption (in contrast, SpatialDE and SPARK are based on Gaussian processes). This is not necessarily a contra: in fact, real data are often complex and not compatible with the theoretical assumptions of some modeling approaches. In the particular task of SVGs detection, I personally believe that it applies the rule “the simpler, the better”: simple ideas sometimes reveal to be useful and efficient for reaching the declared scope, especially when they are also computationally scalable.

At the current status, I believe that this work has laid a promising foundation, but my impression is that it requires some relevant extensions and further evaluations. Sometimes I feel like the work is still at an early stage and it is all but finished. The writing quality is globally acceptable but some parts lack details and precision, especially the Methods Section. If the editorial requirements do not allow the authors to add all the details into the main manuscript, I suggest adding details into a supplementary document.

Methods

Here some comments that might help figuring out the methodological directions that, to me, should be explored:

- According to the proposed model formulation, a gene j is classified as spatially variable if its quantity $r_{D_1 D_2}^{(j)}$ is larger than a certain threshold. This quantity is the ratio between the variances of the local means computed using two different radii, D_1 and D_2 . The authors set $D_1 = 1$ and $D_2 = 3$ (lines 455-456); however, not only these values need to be substantially motivated, but also it is important to test the performance of the method using different radius values, especially of D_2 . Svensson et al. (2018) and the more recent work of Weber et al. (2022) (which actually has not been cited in the literature review) showed that different genes are spatially expressed in space according to different scales. My concern is that, without moving the value of D_2 , you would end up discovering SVGs only within a certain scale. I would thus embed this aspect into the theoretical framework of the model and extend the algorithm in order to select the SVGs after considering many patches. Then, I would integrate this aspect into the simulations. Maybe I'm wrong and $D_2 = 3$ is enough. If so, the authors must show it.
- The paragraph 4.1.3 must be substantially extended as it severely lacks details. The idea of fitting a beta distribution on the values $\{r_{D_1 D_2}^{(j)}\}_{j=1, \dots, N}$ is not immediate to me and must be justified. Furthermore, is it really necessary to use the quantile of a beta? Why not just the empirical quantile?

There is a typo in line 492: I think it should be $j \in [1, N]$.

Lines 495-496 must be explained. What is a situation where the number of null genes is insufficient? How to perform such random generation from the null distribution? What is this "null distribution"?

Please, rewrite.

Simulations

- I agree with the fact that the SPARK framework is a good starting point for the simulations; however, I do believe also that the level of complexity must be increased. Still in the spatial transcriptomic data analysis framework, Sottosanti and Risso (2022) proposed several simulations of growing complexity to test their clustering model. In particular, their simulation n. 4 adds a global level of contamination (which is something different and more complex than a nugget effect) to show that their model is still able to detect the clusters. I know that the purpose behind Sottosanti and Risso' work is different from the one treated here; this is just an example of how the simulation scenarios could be complicated to test the robustness of a method. They also consider genes that are spatially expressed only within some specific areas and not over the whole tissue. This is something that can be considered here as well.

In my personal opinion, some already published papers lack investigation of the robustness of their methods when some assumptions are not respected (as it is for example when a spatial process is nonstationary). If BSP wants to become a new reference point in the analysis of SVGs, the authors need to make a relevant step forward, showing that their method does substantially better than the competitive models in different situations.

- The authors should also include the results obtained applying the nnSVG method of Weber et al. (2022). Even though it's still a pre-print, their results are widely promising, especially if considering the reduced computational cost. In addition, as I already mentioned before, they allow their model to estimate a spatial scale for each and every gene, thus recognizing different types of spatial distributions.
- The authors must write down all the simulation models employed and avoid reporting greek letters and symbols without any contextualization.

Additional Bibliography

Sottosanti, A. and Risso, D. (2022). Co-clustering of spatially resolved transcriptomic data. *The Annals of Applied Statistics*. In press.

<https://www.e-publications.org/ims/submission/AOAS/user/submissionFile/53524?confirm=fb285e6b>

Weber, L. M., Saha, A., Datta, A., Hansen, K. D., & Hicks, S. C. (2022). nnSVG: scalable identification of spatially variable genes using nearest-neighbor Gaussian processes. *bioRxiv*, 2022-05. <https://www.biorxiv.org/content/10.1101/2022.05.16.492124.abstract>

Reviewer #2 (Remarks to the Author):

Major comments:

This work provides an interesting way to analyze the 2D and 3D spatial transcriptomics data and to statistically identify spatially variable genes. Meanwhile, this work is not typical machine learning research (there are no learnable parameters, no trainable models, and no hypothesis). It identifies genes that show significantly different variances between two pre-defined granularities. This approach, though simple, can be effective. Meanwhile, the effectiveness of this approach can be further demonstrated, and the manuscript potentially can be improved by:

1. The rationale of this approach can be further elaborated. For example, why the ratio of the variances of two different granularities (with a difference of 3 folds) can best capture the spatial variances of genes? Why the ratio follows a beta distribution? Could the authors check whether the true distributions in both simulated and real datasets follow a beta distribution? Whether the performance of the model is associated with the goodness of fit to the beta distribution?
2. To justify the necessity of using a ratio of two granularities, could the authors examine the single granularity approach? For example, could the authors check the spatial variances of genes at a series of granularities (1, 2, 4, 8, 16, 32, and 64 units), and for each granularity, use the chi-square distribution (upper tail) to identify significant genes? In the simulated data, please simulate some genes with local patterns (for example, solid circles of a radius of about 2 and center-to-center distance of about 8), some genes with regional patterns, some genes with global patterns, and the rest genes without patterns. Examine whether this single-granularity approach can identify locally, regionally, and globally variant genes at the corresponding granularity. Then on the same simulated data, examine whether BSP can simultaneously identify locally, regionally, and globally variant genes.
3. Could the authors try this scenario: the expression pattern of a specific gene is: a uniform expression value of 100 on the left half of the slide and 10 on the right side, with the slide size of 1000 x 1000, and $D1 = 1$, $D2 = 3$. The rationale is: if the spatial pattern is significantly larger than the size of the big patch, does BSP still work?
4. Hyper-parameter tuning. The spatial resolutions of different technologies vary dramatically across several magnitudes, from 50 nanometers (single-molecule fluorescence in situ hybridization such as MERFISH and SeqFISH+) to 100 microns (e.g., 10x Visium). The default settings (1 unit is defined as the distance that can capture one nearest neighbor, $D1$ is set to 1 unit, and $D2$ is set to 3 units) in different scenarios represent different physical distances and thus reflect different biological meanings. A systematic fine-tuning of these two key parameters for different types of data can help users to use this tool in their research.
5. The assumption of similar spatial resolutions at all three dimensions is true for STARmap, but not for many other technologies such as MERFISH, for which the z-dimension resolution is several hundred folds lower than x- and y-dimensions. The performance evaluation of the tool in both simulated and real data (MERFISH/MERSCOPE, for example) will help users to understand whether this tool applies to such 3-D data.
6. Evaluating the performance of the 3D/BSP vs 2D/meta-analysis (especially the false discoveries) can be helpful. Could the authors perform simulated studies accordingly?

Minor issues:

1. A gene (MAN1A2) is identified as a spatially variant gene by meta-analysis but not when using the BSP (Fig 5 c). However, this is not obvious in Fig 5 c – it looks like this gene varies both in the x-y plane and in the z-direction. Could the authors to provide some statistical metrics to prove that those that are identified by meta-analysis using 2D stacks but not identified by BSP are false discoveries?
2. The maximum variance is used for normalization (Eq. 5). A potential issue is that outliers (which are common in spatial omics data) will dominate the scale of r_{D1D2} . Simulated studies on the sensitivity to outliers (for example, spiking in some extremely large values) can help understand the robustness of the method. If the max-value-based normalization approach is not robust, the sigmoid transformation of log ratios (of the two variances), a commonly used robust transformation, can be an alternative for noisy data such as spatial omics data.
3. For the beta distribution-based hypothesis test, which adjusted P-values are used? FDR?
4. It might be me – but the following links do not work: https://mailmissouri-581my.sharepoint.com/:f:/g/personal/wangjue_umsystem_edu/EnjH6hbt1ptBjWBzoQhGfPoBhFVV

y and <https://www.starmapresources.com/data>.

Reviewer #3 (Remarks to the Author):

The manuscript by Juexin Wang et al. introduces the method BSP for identifications of spatially variable genes from spatial transcriptome data. BSP adapted the concept of spatial granularity to differentiate the SVGs and non-SVGs. The general methodology design is interesting and novel. The authors compared BSP with other existing tools to show superior performance of BSP. Analyses of published data with BSP revealed new SVGs and novel biological insights. This method could be potentially useful for the community. However, more extensive tests and more thorough discussion are needed to address critical issues and illustrate the pros and cons of the method. Following are my comments and questions.

1. In general, I am convinced that the overall design of BSP should be suitable to capture some SVGs with clear patterns in large spatial domains. However, the readers do expect to see more extensive benchmarking of the method with simulated and real data, especially for the SVGs with different types of distribution patterns and in small domains. Following are just some special cases as examples that need to be tested.
 - a. For the small domains in tissue sections, for example, a tiny yet specially functional domain (let's say 1% of the tissue), would the SVGs be found to the right tail of the beta distribution in Fig. 1c? In other words, how efficient is BSP in identifying the SVGs of infrequent cell types or small tissue domains?
 - b. In another scenario, for the functional domains that are sparsely dispersed within a tissue section, would BSP efficiently identify the SVGs of these small and discrete domains?
2. The author only compared BSP with just a couple of the existing methods. This is far from enough. Other methods should be included for the comparisons with simulated and real data.
3. The authors claimed that BSP is non-parametric. One would anticipate that different patch sizes, for both the small and large patches, should lead to different sensitivity and specificity of BSP. This needs to be tested, again, with simulated and real data.
4. Pattern II in Fig. 2B clearly shows that BSP does not perform as well as SPARK or SpatialDE. The statement in from line 161 to 163 is simply not true. This is related to my first comment. It is very likely that BSP works well with some patterns of SVGs but not with other special patterns. The authors should explicitly show the scenarios in which BSP may not work well via more comprehensive tests with real and simulated data.
5. Data drop-out is a major issue of current spatial transcriptome data. The robustness of BSP to forced drop-outs needs to be tested.
6. Presentations of the SVGs identified by BSP with real data are descriptive. Discussion of the biological relevance of the SVGs was simply based on GO analyses. Such results are barely informative. Most of the GO terms were too general to derive any meaningful biological insights. The authors need to do more thorough and deeper analyses of the SVGs, which should be classified into groups based on their distribution patterns.
7. Overall, in my mind, we are not in an urgent need for another tool for identifying the SVGs. There are already many tools available. New methods like BSP would be more appreciated if they can provide further and deeper information of the SVGs, for example, like I mentioned above, classifications of the SVGs based on their distribution patterns, inference of cell types and tissue domains related to the SVGs, functional associations between different domains of the tissues as indicated by the SVG distributions, etc.

Reviewer 1

The article introduces BSP, a model for detecting spatially variable genes (SVGs) using spatial transcriptomic experiments. The idea is to propose a fast and flexible way to determine SVGs in 2D spatial transcriptomic protocols (e.g., MERFISH, SeqFISH+, 10X Visium) and to fill the gap of missing methods for determining SVGs in 3D protocols, which are becoming more common. I find the idea of using a granularity-based method interesting and promising. I see the BSP model as a non-parametric model that is free from any particular modeling assumption (in contrast, SpatialDE and SPARK are based on Gaussian processes). This is not necessarily a contra: in fact, real data are often complex and not compatible with the theoretical assumptions of some modeling approaches. In the particular task of SVGs detection, I personally believe that it applies the rule “the simpler, the better”: simple ideas sometimes reveal to be useful and efficient for reaching the declared scope, especially when they are also computationally scalable. At the current status, I believe that this work has laid a promising foundation, but my impression is that it requires some relevant extensions and further evaluations. Sometimes I feel like the work is still at an early stage and it is all but finished. The writing quality is globally acceptable but some parts lack details and precision, especially the Methods Section. If the editorial requirements do not allow the authors to add all the details into the main manuscript, I suggest adding details into a supplementary document.

Response: We thank the reviewer for the enthusiastic comments on our approach and greatly appreciate all the constructive suggestions and critiques, which have helped us further improve the quality of our manuscript. We added several extensions and evaluations and included more details in the Methods Sections per request. Our point-by-point responses are as follows.

Methods

Here some comments that might help figuring out the methodological directions that, to me, should be explored:

1. According to the proposed model formulation, a gene j is classified as spatially variable if its quantity r is larger than a certain threshold. This quantity is the (j) $D1$ $D2$ ratio between the variances of the local means computed using two different radii, $D1$ and $D2$. The authors set $D1 = 1$ and $D2 = 3$ (lines 455-456); however, not only these values need to be substantially motivated, but also it is important to test the performance of the method using different radius values, especially of $D2$. Svensson et al. (2018) and the more recent work of Weber et al. (2022) (which actually has not been cited in the literature review) showed that different genes are spatially expressed in space according to different scales. My concern is that, without moving the value of $D2$, you would end up discovering SVGs only within a certain scale. I would thus embed this aspect into the theoretical framework of the model and extend the algorithm in order to select the SVGs after considering many patches. Then, I would integrate this aspect into the simulations. Maybe I'm wrong and $D2 = 3$ is enough. If so, the authors must show it.

Response: We acknowledge the importance of testing the method's performance with different radius values of \$D2\$. As suggested, we have included references to SpaRTaCo by Sottosanti, A. and Risso, D. (2022) and nnSVG by Weber et al. (2022) in the Introduction section.

We have conducted a systematic exploration of parameter selections in the BSP model. In the BSP model, D_1 is normalized to a unit one by estimated density using Eq. (1) for 2D space and Eq. (2) for 3D volume. Once D_1 is fixed to a unit of one, the only parameter that requires selection is D_2 , representing the granularity. Additionally, we have adjusted their strategies in **(A)** the added simulations using three 3D spatial patterns, and **(B)** the real data benchmarks of mouse olfactory bulb and human breast cancer.

(A) In the 3D simulations in all three 3D spatial patterns, including curved stick (Pattern I), thin plate (Pattern II), and irregular lump (Pattern III) described in Section 4.3.1, the power analysis results are presented in the newly added **Supplementary Figure 22** as follows. After checking the impact of various D_2 radius values with 2.0, 2.5, 3.0, 3.5, 4.0, 5.0, and 10.0, we observed that the BSP method exhibits insensitivity to the parameter selection of D_2 within the range of 2 to 5. It is also noteworthy that the statistic power decreases when an excessively large D_2 is used (e.g., $D_2=10$). Based on experimental results across all the simulations and case studies conducted on different SRT platforms and datasets in this work, we uniformly define D_2 as 3, which has proven to yield satisfactory results.

Supplementary Figure 22: Power comparisons with gradient scales of big-small-batch. The power analysis was conducted for a series of D_2 values in the BSP model. Simulations using the continuous 3D pattern I (curved stick), pattern II (thin plate), and pattern III (irregular lump) are shown in the left, middle, and right columns, respectively. The results show very slight differences when D_2 ranges from 2.0 to 5.0.

(B) In the real data benchmarks of mouse olfactory bulb and human breast cancer, we employed a gradient of D_2 values ranging from 2.0 to 10.0 to evaluate the performance of BSP in identifying marker genes as reported in their original studies. The mouse olfactory bulb dataset consists of 10 marker genes, and the human breast cancer dataset includes 14 marker genes (as described in Section 2.3). From the results presented in the newly added **Supplementary Table 36**, we observed that the performances of BSP are basically stable with ranges of D_2 selection in the real data benchmarks. For the mouse olfactory bulb study, BSP consistently identifies more than 8 marker genes when D_2 values range from 2 to 8. Similarly, in human breast cancer study, BSP successfully identifies 13 marker genes when D_2 values range from 2 to 4.

These revisions, findings, and descriptions have been incorporated into Lines 373-379 in Section 2.5 and Lines 573-575 in Section 4.1.1

Section 2.5

Specially, BSP consistently identifies SVGs regardless of selecting different radius values on the scale of the big patch (D_2) in contrast to the small patch (D_1). Following co-clustering strategies¹⁹, we set the small patch as the reference unit with a fixed value one, power analysis reveals slight differences when D_2 ranges from 2.0 to 5.0. The same trends can be observed in all three 3D continuous patterns presented in **Supplementary Figure 22**. By making reasonable choices ($D_2 \leq 5.0$), BSP exhibits insensitivity to parameter selection, enabling its application across a wide range of SRT platforms and datasets.

Section 4.1.1

...More comprehensive exploration shows the BSP performance is insensitive in selecting D_2 on scenarios of simulations (**Supplementary Figure 22**) and on real data of mouse olfactory bulb and human breast cancer studies (**Supplementary Table 36**).

Upon careful analysis, we observed that reasonably selecting D_2 with the range of 2.0 to 5.0 exhibits insensitivity to the power of performance across three 3D patterns. Notably, we uniformly define D_2 as 3 units, and this fixed parameter has proven to work well in all the simulations and case studies conducted across various SRT platforms and datasets in this study. This result suggests that BSP as a data-driven and non-parametric model is robust and effective.

2. The paragraph 4.1.3 must be substantially extended as it severely lacks details. The idea of fitting a beta distribution on the values $\{r_j\}_{j=1, \dots, N}$ is not immediate to me and must be justified. Furthermore, is it really necessary to use the quantile of a beta? Why not just the empirical quantile? There is a typo in line 492: I think it should be $j \in [1, N]$. Lines 495-496 must be explained. What is a situation where the number of null genes is insufficient? How to perform such random generation from the null distribution? What is this "null distribution"? Please, rewrite.

Response: Thanks for your constructive suggestion, we corrected the typo and rewrote Section 4.1.3.

As suggested, we conducted an extensive analysis to test the fitness of a serial of different statistical distributions using two approaches: **(A)** Q-Q plot and **(B)** goodness-of-fit test with Cramer-von Mises criterion using i) simulations on three continuous 3D patterns, ii) mouse olfactory bulb data, and iii) human breast cancer data.

In the simulations involving three 3D spatial patterns, 900 SVGs were constructed to present the spatial pattern, while 9,000 non-SVGs were generated through location permutations. Each simulation was repeated 10 times to ensure the robustness of the results.

(A) Through our analysis, we found that the lognormal distribution better fits the test scores obtained from BSP in both simulated and real datasets (newly added **Supplementary Figures 39**). Specifically, the lognormal distribution demonstrates a good fit for the test scores from simulation and mouse olfactory bulb data. Although there are some deviations on the tail of the Q-Q plot for the human breast cancer data, the lognormal distribution still performs reasonably well. Conversely, the beta distribution (newly added **Supplementary Figure 40**) fits well with the simulation data but does not provide a satisfactory fit for the real datasets of mouse olfactory bulb and human breast cancer.

(B) Moreover, we conducted the goodness-of-fit test with the Cramer-von Mises criterion on the simulated datasets. Due to computational limitations, we examined the p-values for every thousand non-SVGs and presented the results in newly added **Supplementary Tables 34-35**. From the simulations on three 3D spatial patterns, most p-values were greater than 0.05 when D2 was selected within the reasonable range (specifically, from 2 to 5 as demonstrated in **Supplementary Figure 22** in response to critique #1, indicating no statistically significant differences between the distribution of test scores and the fitted lognormal distribution. Based on these findings, we decided to expand the choices of statistical distributions in the BSP model in the revision, with the lognormal distribution set as the default choice while still offering the option of using the beta distribution. Consequently, we have updated most of the simulations and experiments in BSP to utilize the lognormal distributions.

We sincerely appreciate the reviewer's excellent suggestion. The revised BSP with lognormal distribution has shown even better results in the benchmarks of mouse olfactory bulb and human breast cancer datasets. In the case of the 10 marker genes from the original mouse olfactory bulb study, the previous BSP with a beta distribution identified 8 out of 10 marker genes missing genes *Nmb* and *Sv2b*, while the revised BSP with lognormal distribution detected 9 out of 10 marker genes, only missing *Sv2b* gene with a marginal p -value 0.0518. Similarly, for the 14 marker genes from the original human breast cancer study, the previous BSP with beta distribution detected 12 out of 14 marker genes missing genes *PEG10* and *PIP*, whereas the revised BSP with lognormal distribution detected 13 of 14 marker genes, only missing *PIP* gene.

We incorporated the related description and revisions to Lines 201-216 in Section 2.3, Lines 600-613 in Section 4.1.3, and Lines 526-530 in the Discussion of the manuscript.

Section 2.3

*The mouse olfactory bulb dataset contains 11,274 genes measured on 260 spots using SRT sequencing. BSP detected 9 of 10 marker genes from the original study¹¹, while SpatialDE detected 3, SPARK detected 8, nnSVG detected 6, and SPARK-X detected 0. **Figure 3A** and **Supplementary Figure 4** show the comparison between different methods. The only marker gene BSP missed was *Sv2b*, with p -values of 0.0518. We reason this missed marker gene has expression variances confined to many isolated, relatively small regions, which could result in the same variances in both big and small patches (**Supplementary Figure 5**). The human Breast cancer dataset contains 5,262 genes measured on 250 spots by SRT sequencing. BSP detected 13 of 14 marker genes identified as SVGs from the original study, while SpatialDE detected 7, SPARK detected 10, and SPARK-X detected 8. The result comparison is shown in **Figure 3B** and **Supplementary Figure 6**. The marker gene BSP missed was *PIP* with p -values of 0.0850. The other two FISH-based datasets include the hippocampus dataset, consisting of 249 genes on 131 cells obtained by SeqFISH, and the mouse hypothalamus preoptic region composed of 160 genes on 257 cells by MERFISH. BSP identified most of the marker genes reported in the original studies. Detailed results for mouse olfactory bulb, human breast cancer, hippocampus, and hypothalamus preoptic regions are provided in **Supplementary Tables 2-5**, respectively.*

Section 4.1.3

After $r_{D_1 D_2}^{(j)}$ is calculated with all the genes, $j \in [1, N]$, a lognormal distribution is approximated for the distribution of $r_{D_1 D_2}^{(j)}$ using the stat packages from sklearn⁵⁵. Depending on the

characteristics of the data, a beta distribution is considered as an alternative approximation for $r_{D_1 D_2}^{(j)}$ using the same packages. To tolerate the potential noise and long-tail deviations, a significance p-value is assigned to each gene if $r_{D_1 D_2}^{(j)}$ exceeds the upper tail of the fitted distribution at a significance level of $100 * (1 - \alpha)\%$, where α refers to the significance level (usually set as 0.05). Our hypothesis is that only a small portion of genes located in the tail of the distribution are SVGs, while the majority of genes are non-SVGs that are spatially independent. This hypothesis is particularly applicable in high-throughput SRT platforms like 10X Visium, which involve more than thousands of genes. However, in low-throughput SRT platforms such as FISH, where there are insufficient genes (hundreds or even less), a set of random genes permuted across spatial locations as null genes are generated to compensate for non-SVGs. These null genes help estimate the complete distribution for practical usage.

Simulations

1. I agree with the fact that the SPARK framework is a good starting point for the simulations; however, I do believe also that the level of complexity must be increased. Still in the spatial transcriptomic data analysis framework, Sottosanti and Risso (2022) proposed several simulations of growing complexity to test their clustering model. In particular, their simulation n. 4 adds a global level of contamination (which is something different and more complex than a nugget effect) to show that their model is still able to detect the clusters. I know that the purpose behind Sottosanti and Risso' work is different from the one treated here; this is just an example of how the simulation scenarios could be complicated to test the robustness of a method. They also consider genes that are spatially expressed only within some specific areas and not over the whole tissue. This is something that can be considered here as well. In my personal opinion, some already published papers lack investigation of the robustness of their methods when some assumptions are not respected (as it is for example when a spatial process is nonstationary). If BSP wants to become a new reference point in the analysis of SVGs, the authors need to make a relevant step forward, showing that their method does substantially better than the competitive models in different situations.

Response: Thanks for your suggestion. We have increased the level of complexity of the simulations by adding 5 additional simulations, including (1) simulation with dropout issues, (2) comparison between 3D modeling and meta-analysis on 2D slices, (3) performance on discrete and continuous spatial patterns, (4) performances on 3D simulations within inconsistent within-plane and inter-plane resolution, and (5) performance comparisons with gradient scales of big-small-batch. See more details in Lines 344-381 in Section 2.5:

Experiment 1: Simulations with dropout issues

*We further investigated the capability of BSP on SRT in the presence of dropout issues⁴⁴, where only a subset of the transcriptome is captured by sequencing due to technical limitations. Dropout effects were simulated by randomly assigning zero-values to a certain proportion (10%, 20%, and 30%) of cells in continuous patterns I, II, and III with moderate pattern size, signal strength, and noise level. The power analysis (**Supplementary Figure 19**) indicates the robustness of BSP in handling SRT with a reasonable dropout rate.*

Supplementary Figure 19: Power comparisons with varying dropout rates. Simulations were performed using a fixed moderate pattern size, moderate signal strength, and moderate noise level. In these nine power charts, simulations with low (10%), moderate (20%), and high (30%) dropout rates are in the left, middle, and right columns, respectively. Simulations using the continuous 3D Pattern I (curved stick), Pattern II (thin plate), and Pattern III (irregular lump) are shown in the top, middle, and bottom rows, respectively.

Experiment 2: Comparison between 3D modeling and meta-analysis on 2D slices

To demonstrate the advantages of utilizing 3D SRT, we compared SVGs identification performances between direct detection from 3D data and detection through meta-analysis,

which combined individual analyses on each 2D slice using various combined probability tests. We performed simulations on all continuous patterns I, II, and III with moderate pattern size, signal strength, and noise level (**Figure 4C**). Our findings indicate that employing the BSP model on the 3D data yields superior power than the meta-analysis conducted on the results from 2D slices.

Figure 4C) Power comparisons between 3D SRT and meta-analysis on 2D slices. Simulations were performed using a fixed moderate pattern size, moderate signal strength, and moderate noise level. The red line represents the p-value from BSP modeling the 3D SRT. Other colored lines depict p-values from various meta-analysis provided by the SciPy package, including Fisher, Pearson, Tippett, Stouffer, and Mudholkar, applied to BSP on 2D slices. Simulations using the 3D pattern I (curved stick), pattern II (thin plate), and pattern III (irregular lump) are shown in the left, middle, and right columns, respectively.

Experiment 3: Performances on discrete and continuous spatial patterns

In addition to examining continuous patterns with global influence (depicted by the three 3D patterns in **Figure 4A**), we also evaluate the model performances on local discrete spatial patterns (**Figure 4D**). These patterns often manifest in isolated, small tissue domains in practical scenarios. Power analysis shows that the BSP model can accurately identify SVGs associated with these local discrete spatial patterns (**Figure 4E**). **Supplementary Figure 20** represents a comprehensive power analysis for simulations involving varying pattern sizes, signal strength, and noise levels.

The simulation details are added to Lines 669-686 in Methods Section 4.3.2.

Section 4.3.2 3D simulation on FISH data with discrete spatial patterns

To further evaluate the model's performance on data with locally influential discrete patterns (**Figure 4D**), we designed a set of 3D simulation scenarios within the ranges of x- and y-coordinates from 0 to 30. The local discrete spatial patterns were constructed using solid spheres with a center-to-center distance of 8 units. Specifically, sixteen center points were selected, with fixed z-coordinates of 5.5, and the x- and y- coordinates were generated from a sequence of numbers from 3 to 27 with an interval of 8. To introduce randomness into spatial patterns, we incorporated a uniformly distributed random variable ranging from -2 to 2 and added it to the coordinates of each center point. The cells within the spheres were marked, and expression values were assigned as previously described.

In these simulations, SVGs were generated using the Irregular lump pattern described in 4.3.1. We created 500 SVGs with locally influential discrete patterns and an additional 500 SVGs with globally influential continuous patterns, along with 9,000 non-SVGs as permuted genes without any spatial pattern. We considered three scenarios to compare the effects of pattern sizes, signal strengths, and noise levels, respectively. We adjusted radius values from 1.5 to 2.0 (small and moderate), expected expression fold changes from 2 to 2.5 (small and moderate), and noise levels from 2.0 to 3.0 (moderate and high) as previously described. To ensure robustness, ten replicates were generated to perform the power analysis.

Figure 4D) Example of simulated discrete spatial patterns in isolated local domains. **E)** Power analysis for identifying SVGs with mixed discrete and continuous patterns. This simulation contains 500 SVGs with discrete patterns, 500 SVGs with continuous patterns, and 9,000 non-SVGs.

Supplementary Figure 20: Performances on discrete and continuous spatial patterns. This simulation contains 500 SVGs with discrete patterns, 500 SVGs with continuous patterns, and 9,000 non-SVGs. Simulations were conducted with varied pattern sizes (small to moderate, left to right), signal strength

(small to moderate, left to right), and noise level (moderate to high, left to right) while holding the rest two parameters (top to bottom).

Experiment 4: Performances on 3D simulations within inconsistent within-plane and inter-plane resolution

*In current 3D SRT, especially with data obtained from FISH-based sequencing techniques, it is common for the inter-plane spatial resolution (z-axis) to be considerably lower than the within-plane resolution (x- and y- axes)^{4,6}. To further assess the model's performance when the assumption of similar or equivalent spatial resolutions across all three dimensions is violated, we generated 3D simulations with varying scales of spatial resolutions between the within-plane (x- and y- axes) and inter-plane (z-axis) dimensions. Power analysis indicates that the BSP model accurately identifies SVGs in 3D SRT, even in scenarios with discrepancies between the inter-plane spatial and within-plane resolutions (**Figure 4F**). **Supplementary Figure 21** details a comprehensive power analysis on simulations with varying pattern sizes, signal strength, and noise levels.*

The simulation details are added to Lines 687-698 in Methods Section 4.3.3.

4.3.3 3D simulation on FISH data with varying spatial resolutions in the z-axis

*In practical scenarios, the assumption of similar or equivalent spatial resolutions across all three dimensions may not hold, particularly in FISH-based sequencing techniques where the inter-plane spatial resolution (z-axis) is often significantly lower than the within-plane resolution (x- and y- axes). To systematically test the capability of BSP on 3D spaces, we increased the resolution of the z-axis by multiplying the z-coordinates by ten ($z = 10, 20, \dots, 100$) and replicated the simulation as described in Section 4.3.1. Considering the substantial differences between the z-coordinate and x- or y- coordinates, we fixed the direction of random walks in the z-axis to ensure that the sliced planes captured the simulated spatial pattern (**Supplementary Figure 42**). We varied the pattern sizes (small and moderate radius = 1.5 or 2.0), signal strengths (small and moderate, fold change = 2.0 or 2.5), and noise levels (small and moderate, $\sigma=1$ or 2), respectively. The performance of the BSP model was compared with SPARK-X.*

Figure 4F) Power analysis for identifying SVGs in 3D simulations with inconsistent within-plane and inter-plane resolution.

Supplementary Figure 21: Performances on 3D simulations within inconsistent within-plane and inter-plane resolution. Simulations were conducted with varied pattern sizes (small to moderate, left to right), signal strength (low to moderate, left to right), and noise level (moderate to high, left to right) while holding the rest two parameters as moderate (top to bottom).

Supplementary Figure 42: 3D simulation strategies. a) **Simulation adopted:** Fixed the direction of the random walks in the z-axis to ensure that the sliced planes can capture the simulated spatial pattern. b) **Simulation avoided:** The direction of the random walks may be horizontal with the z-axis, so the simulated 3D spatial pattern cannot be captured by the sliced planes.

Experiment 5: Performance comparisons with gradient scales of big-small-batch

Please refer to the response to critique #1 for this experiment.

2. The authors should also include the results obtained applying the nnSVG method of Weber et al. (2022). Even though it's still a pre-print, their results are widely promising, especially if considering the reduced computational cost. In addition, as I already mentioned before, they allow their model to estimate a spatial scale for each and every gene, thus recognizing different types of spatial distributions.

Response: Thank you for the suggestion. In the revision, we included the nnSVG method in all the analyses and other competitive methods such as spatialDE, SPARK, SPARK-X, and Moran's I. These methods have been used in 2D simulations with varying pattern sizes, signal strengths, and noises, as well as in real data studies. Our comprehensive evaluation of 2D simulations and benchmarks using mouse olfactory bulb data and human breast cancer data shows that BSP is accurate and fast compared to the competitive methods. On 3D simulations, BSP demonstrates superior accuracy compared to other methods.

(A) In performance comparisons on simulations, **Figure 2B** in the following shows the power comparison among different methods using moderate pattern size, noise level, and signal strength for the spatial expression patterns I, II, and III (left to right) in the simulations. To provide a more comprehensive power analysis, we have included additional comparisons considering different signal-to-noise ratios

(weak, moderate, and strong) with a moderate noise level in the newly added **Supplementary Figure 1**. Moreover, we have examined different noise levels (low, moderate, high) with a moderate signal-to-noise ratio, as shown in the newly added **Supplementary Figure 2**.

We also conducted computational time and memory usage comparisons with other competitive methods on an Ubuntu 16.04.4 LTS workstation with Intel(R) Xeon(R) W-2125 CPU @ 4.00GHz and 32 GB memory in the newly added **Supplementary Table 1**. **Figure 2C** in the following shows computational time costs with an increasing number from 2,000 genes to 20,000 genes and fixed 2,000 spots. **Figure 2D** shows the computational time costs with an increasing number from 1,000 spots to 10,000 spots and fixed 10,000 genes. The corresponding memory usage is shown in the newly added **Supplementary Figure 3**.

The updated analysis and benchmarking have been included in the revised manuscript in Lines 152-192 in Results Section 2.2 and corresponding Figure 2, Supplementary Figures 1-3, Supplementary Table 1.

Figure 2: Power analysis for 2D simulation. **A)** Spatial expression patterns I-III (left to right) as defined in SpatialDE and SPARK. **B)** Power comparison among the different methods with moderate noise level, and moderate signal strengths for the spatial expression patterns I, II, and III (left to right). All simulations were generated based on the mouse olfactory bulb data with 260 spots of cells. Each simulation replicate contains 1,000 SVGs and 9,000 non-SVGs. **C)** Computational time costs with an increasing number of genes and fixed 2,000 spots. The y-axis indicates the logarithmic of the running time in seconds. **D)** Computational time costs with an increasing number of spots and fixed 10,000 genes. The y-axis indicates the logarithmic of the running time in seconds. The time cost greater than 48 hours are not shown in the figure.

(B) We also tested the performance of nnSVG on real data benchmarks on mouse olfactory bulb. Unfortunately, the R package of nnSVG continuously throws errors on the human breast cancer dataset. We raised the issue on the nnSVG Github page <https://github.com/lmweber/nnSVG/issues/16>, but have not received any response yet. The updated analysis and benchmarking have been included in the revised manuscript Lines 201-207 in Results Section 2.3, corresponding to Figure 3A, and Supplementary Figure 4.

Section 2.3

*The mouse olfactory bulb dataset contains 11,274 genes measured on 260 spots using SRT sequencing. BSP detected 9 of 10 marker genes from the original study¹¹, while SpatialDE detected 3, SPARK detected 8, nnSVG detected 6, and SPARK-X detected 0. **Figure 3A** and **Supplementary Figure 4** show the comparison between different methods. The only marker gene BSP missed was Sv2b, with p-values of 0.0518. We reason this missed marker gene has expression variances confined to many isolated, relatively small regions, which could result in the same variances in both big and small patches (**Supplementary Figure 5**).*

Figure 3. SVGs identified by BSP in biological analysis. A) Number of marker genes identified with different methods in mouse olfactory bulb study (the original study identified ten marker genes). Intersections between genes identified from different methods were included in the analyses.

Supplementary Figure 4. Venn diagram of marker genes identified by BSP, nnSVG, SPARK, and SpatialDE in mouse olfactory bulb research. The original study includes 10 marker genes. The result of SPARKX is not shown for it identified 0 out of 10 marker genes.

3. The authors must write down all the simulation models employed and avoid reporting greek letters and symbols without any contextualization.

Response: We carefully checked the manuscript and added explanations and annotations for all the Greek letters and symbols in the methods sections. In other parts of the manuscript, we have opted to describe spatial patterns as small, moderate, and large, signal strength as weak, moderate, and high, and noise as low, moderate, and high, to avoid explicitly using Greek letters and symbols. These definitions are provided in Lines 665-667 in Methods Section 4.3.1.

...In the 3D simulations, we varied the pattern sizes ($r = 1.5, 2.0, 2.5$), expression fold changes ($FC = 2.0, 2.5, 3.0$), and noise levels ($\tau = 0, 1, 2$) across continuous spatial patterns I, II, and III...

Additional Bibliography

Sottosanti, A. and Risso, D. (2022). Co-clustering of spatially resolved transcriptomic data. The Annals of Applied Statistics. In press. <https://www.e-publications.org/ims/submission/AOAS/user/submissionFile/53524?confirm=fb285e6b> Weber, L. M., Saha, A., Datta, A., Hansen, K. D., & Hicks, S. C. (2022). nnSVG: scalable identification of spatially variable genes using nearest-neighbor Gaussian processes. bioRxiv, 2022-05. <https://www.biorxiv.org/content/10.1101/2022.05.16.492124.abstract>

Response: We have added these references in our revision, including the updated nnSVG reference.

Reviewer 2:

Major comments:

This work provides an interesting way to analyze the 2D and 3D spatial transcriptomics data and to statistically identify spatially variable genes. Meanwhile, this work is not typical machine learning research (there are no learnable parameters, no trainable models, and no hypothesis). It identifies genes that show significantly different variances between two pre-defined granularities. This approach, though simple, can be effective. Meanwhile, the effectiveness of this approach can be further demonstrated, and the manuscript potentially can be improved by:

Response: We thank the reviewer for the positive comments and value the following constructive suggestions. Please find our point-by-point responses below.

1. The rationale of this approach can be further elaborated. For example, why the ratio of the variances of two different granularities (with a difference of 3 folds) can best capture the spatial variances of genes? Why the ratio follows a beta distribution? Could the authors check whether the true distributions in both simulated and real datasets follow a beta distribution? Whether the performance of the model is associated with the goodness of fit to the beta distribution?

Response: Conceptually, the rationale of the BSP method is based on utilizing the paired two different granularities to analyze the spatial space. The underlying hypothesis is that the spatial patterns of SVGs are more conserved under different granularities than non-SVGs. **Figure 1** illustrates the schema of BSP. Our investigations found that the ratio between variances of big and small patches can effectively capture certain spatial characteristics exhibited by SVGs. We discussed the idea in Lines 505-508 of the Discussion Section, *“Intuitively, the big patch provides a global view of the spatial pattern with a lower resolution, while the small patch focuses on the local details with a higher resolution. Using ratios between variances of the paired patches, BSP can accurately delineate the spatial patterns in a quantitative manner.”* Then we used 2D & 3D simulations and real data to demonstrate the power of the BSP model.

(I) Performance comparisons with gradient scales of big-small-batch

To demonstrate the ratio of the variances of two different granularities (with a difference of 3 folds) can best capture the spatial variances of genes, we designed comprehensive experiments to test model performances using different radii of D2 in **(A)** the additional simulations using three 3D spatial patterns, and **(B)** the real data benchmarks of mouse olfactory bulb and human breast cancer.

(A) In the 3D simulations in all three 3D spatial patterns, including curved stick (Pattern I), thin plate (Pattern II), and irregular lump (Pattern III) described in Section 4.3.1, the power analysis results are presented in the newly added **Supplementary Figure 22**. After checking the impact of various D2 radius values with 2.0, 2.5, 3.0, 3.5, 4.0, 5.0, and 10.0, we observed that the BSP method exhibits insensitivity to the parameter selection of D2 within the range of 2 to 5. It is also noteworthy that the statistic power decreases when an excessively large D2 is used (e.g., D2=10). Based on experimental results across all the simulations and real data studies conducted on different SRT platforms and datasets in this work, we uniformly define D2 as 3, which has proven to yield satisfactory results.

(B) In the real data benchmarks of mouse olfactory bulb and human breast cancer, we employed a gradient of D2 values ranging from 2.0 to 10.0 to evaluate the performance of BSP in identifying marker genes as reported in their original studies. The mouse olfactory bulb dataset consists of 10 marker genes, and the human breast cancer dataset includes 14 marker genes (as described in Section 2.3). From the results presented in the newly added **Supplementary Table 36**, we observed that the performances of BSP are basically stable with ranges of D2 selection in the real data benchmarks. For the mouse olfactory bulb study, BSP consistently identifies more than 8 marker genes when D2 values range from 2 to 8. Similarly, in human breast cancer study, BSP successfully identifies 13 marker genes when D2 values range from 2 to 4.

These revisions, findings, and descriptions have been incorporated into Lines 373-379 in Section 2.5 and Lines 572-575 in Section 4.1.1

(II) Exploration of selecting appropriate statistical distribution

To answer the question related to beta distribution, we conducted an extensive analysis to test the fitness of a series of different statistical distributions using two approaches: **(A)** Q-Q plot and **(B)** goodness-of-fit test with Cramer-von Mises criterion on i) simulations on three continuous 3D patterns, ii) mouse olfactory bulb data, and iii) human breast cancer data.

In the simulations involving three 3D spatial patterns, 900 SVGs were constructed to present the spatial pattern, while 9,000 non-SVGs were generated through location permutations. Each simulation was repeated 10 times to ensure the robustness of the results.

(A) Through our analysis, we found that the lognormal distribution better fits the test scores obtained from BSP in both simulated and real datasets (newly added **Supplementary Figures 39**). Specifically, the lognormal distribution demonstrates a good fit for the test scores from simulation and mouse olfactory bulb data. Although there are some deviations on the tail of the Q-Q plot for the human breast cancer data, the lognormal distribution still performs reasonably well. Conversely, the beta distribution (newly added **Supplementary Figure 40**) fits well with the simulation data but does not provide a satisfactory fit for the real datasets of mouse olfactory bulb and human breast cancer.

(B) Moreover, we conducted the goodness-of-fit test with the Cramer-von Mises criterion on the simulated datasets. Due to computational limitations, we examined the p-values for every thousand non-SVGs and presented the results in the newly added **Supplementary Tables 34-35**. From the simulations on three 3D spatial patterns, most p-values were greater than 0.05 when D2 was selected within the reasonable range (specifically, from 2 to 5 as demonstrated in Supplementary Figure 22 in **response to critique #1**, indicating no statistically significant differences between the distribution of test scores and the fitted lognormal distribution. Based on these findings, we decided to expand the choices of statistical distributions in the BSP model in the revision, with the lognormal distribution set as the default choice while still offering the option of using the beta distribution. Consequently, we have updated most of the simulations and experiments in BSP to utilize the lognormal distributions.

Notably, we sincerely appreciate the reviewer's suggestion. The revised BSP with lognormal distribution has shown even better results in the benchmarks of mouse olfactory bulb and human breast cancer datasets. In the case of the 10 marker genes from the original mouse olfactory bulb study, the previous

BSP with beta distribution identified 8 out of 10 marker genes missing genes *Nmb* and *Sv2b*, while the revised BSP with lognormal distribution detected 9 out of 10 marker genes, only missing *Sv2b* gene with a marginal p -value 0.0518. Similarly, for the 14 marker genes from the original human breast cancer study, the previous BSP with beta distribution detected 12 out of 14 marker genes missing genes *PEG10* and *PIP*, whereas the revised BSP with lognormal distribution detected 13 of 14 marker genes, only missing *PIP* gene.

We incorporated the related description and revisions to Lines 201-216 in Section 2.3, Lines 600-613 in Section 4.1.3, and Lines 526-530 in the Discussion of the manuscript.

2. To justify the necessity of using a ratio of two granularities, could the authors examine the single granularity approach? For example, could the authors check the spatial variances of genes at a series of granularities (1, 2, 4, 8, 16, 32, and 64 units), and for each granularity, use the chi-square distribution (upper tail) to identify significant genes? In the simulated data, please simulate some genes with local patterns (for example, solid circles of a radius of about 2 and center-to-center distance of about 8), some genes with regional patterns, some genes with global patterns, and the rest genes without patterns. Examine whether this single-granularity approach can identify locally, regionally, and globally variant genes at the corresponding granularity. Then on the same simulated data, examine whether BSP can simultaneously identify locally, regionally, and globally variant genes.

Response: This is a very interesting idea. We conducted several extended experiments to test this idea in the revision.

(A) As suggested, we added a series of simulations to examine the performances of a single granularity. In these simulations, the spatial variances of genes were calculated at various granularities (1.0, 2.0, 2.5, 3.0, 3.5, 4.0, 5.0, 8.0, 10.0, 16.0, excluding 32 and 64 due to limited range of coordinates) on the simulated 3D datasets with moderate pattern size, signal strength, and noise level. These simulations encompassed three distinct 3D patterns described in Section 4.3.1, namely curved stick (Pattern I), thin plate (Pattern II), and irregular lump (Pattern III). **We fitted a chi-square distribution for each granularity and determined the p-value for each gene.** We observed that the obtained p -values for genes with and without spatial patterns were very similar, indicating a poor capacity to distinguish the SVGs from non-SVGs. To address this issue, we calculated **calibrated p-value** using Stouffer's method to enhance such differentiation. Nonetheless, the histogram of the calibrated p -values on the simulation (specifically, the first simulated dataset with Pattern I, similar distributions were observed for the other patterns) revealed poor capacity in distinguishing SVGs and non-SVGs (**Extended Figure 1** as follows). To further compare the results, we conducted a power analysis, which highlighted significant gaps between the single granularity-based approach (Calibrated p -value in red line) and the BSP approach with the paired big-small patch (Default in blue line) (**Extended Figure 2** as follows).

The main reason behind this disparity is that BSP measures the **velocity of changes** in the variances of local means as the radius changes, which is captured by the ratios between the variances using different granularities. In contrast, the single granularity approach solely focuses on the variances themselves, measured at a single granularity. Both SVGs and non-SVGs may have a significant variance at a certain granularity, but an SVG's variance is often more consistent than non-SVGs across different granularities (see Figure 1). Hence, a single-granularity approach does not work well for identifying SVGs, but change of variance at two granularities works well as this paper has demonstrated.

Extended Figure 1. The histogram of the calibrated p-values on the 3D simulations with Pattern I (curved stick). Similar trends were observed in simulations with other patterns.

Extended Figure 2. Performance comparisons between the single granularity approach and the BSP approach. Simulations using the continuous 3D pattern I (curved stick), pattern II (thin plate), and pattern III (irregular lump) are shown in the left, middle, and right columns, respectively. The red line represents the power of the single granularity approach (Calibrated p-value). The blue line represents the power of the proposed BSP approach with paired big-small patch (Default).

(B) As suggested, we conducted experiments to evaluate the performance of the BSP method in identifying locally, regionally, and globally variant genes simultaneously. Power analysis indicates the versatility and effectiveness of BSP in capturing spatial gene expression patterns across different contexts in **Figures 4D, 4E,** and **Supplementary Figure 20.** We added the paragraph in Lines 356-362 in Results Section 2.5 and Lines 669-686 in Methods Section 4.3.2.

Section 2.5

In addition to examining continuous patterns with global influence (depicted by the three 3D patterns in **Figure 4A**), we also evaluate the model performances on local discrete spatial patterns (**Figure 4D**). These patterns often manifest in isolated, small tissue domains in practical scenarios. Power analysis shows that the BSP model can accurately identify SVGs associated with these local discrete spatial patterns (**Figure 4E**). **Supplementary Figure 20** represents a comprehensive power analysis for simulations involving varying pattern sizes, signal strength, and noise levels.

Section 4.3.2 3D simulation on FISH data with discrete spatial patterns

To further evaluate the model's performance on data with locally influential discrete patterns (**Figure 4D**), we designed a set of 3D simulation scenarios within the ranges of x - and y -coordinates from 0 to 30. The local discrete spatial patterns were constructed using solid spheres with a center-to-center distance of 8 units. Specifically, sixteen center points were selected, with fixed z -coordinates of 5.5, and the x - and y -coordinates were generated from a sequence of numbers from 3 to 27 with an interval of 8. To introduce randomness into spatial patterns, we incorporated a uniformly distributed random variable ranging from -2 to 2 and added it to the coordinates of each center point. The cells within the spheres were marked, and expression values were assigned as previously described.

In these simulations, SVGs were generated using the Irregular lump pattern described in 4.3.1. We created 500 SVGs with locally influential discrete patterns and an additional 500 SVGs with globally influential continuous patterns, along with 9,000 non-SVGs as permuted genes without any spatial pattern. We considered three scenarios to compare the effects of pattern sizes, signal strengths, and noise levels, respectively. We adjusted radius values from 1.5 to 2.0 (small and moderate), expected expression fold changes from 2 to 2.5 (small and moderate), and noise levels from 2.0 to 3.0 (moderate and high) as previously described. To ensure robustness, ten replicates were generated to perform the power analysis.

Figure 4D) Example of simulated discrete spatial patterns in isolated locate domains. **E)** Power analysis for identifying SVGs with mixed discrete and continuous patterns. This simulation contains 500 SVGs with discrete patterns, 500 SVGs with continuous patterns, and 9,000 non-SVGs.

Supplementary Figure 20: Performances on discrete and continuous spatial patterns. This simulation contains 500 SVGs with discrete patterns, 500 SVGs with continuous patterns, and 9,000 non-SVGs. Simulations were conducted with varied pattern sizes (small to moderate, left to right), signal strength

(small to moderate, left to right), and noise level (moderate to high, left to right) while holding the rest two parameters (top to bottom).

3. Could the authors try this scenario: the expression pattern of a specific gene is: a uniform expression value of 100 on the left half of the slide and 10 on the right side, with the slide size of 1000 x 1000, and $D1 = 1$, $D2 = 3$. The rationale is: if the spatial pattern is significantly larger than the size of the big patch, does BSP still work?

Response: We constructed the simulations per the reviewer's request accordingly. However, we encountered challenges due to the large size of the simulated data with a slide size of 1,000*1,000. This size exceeded the capacity for processing, even surpassing the largest available HDST olfactory bulb data used in our study, which consists of 19,950 genes measured on 181,367 spots, as described in Lines 281-282 in Section 2.4.

To address this limitation, we created a smaller simulation with a slide size of 300 x 300. In this simulation, the expression pattern of a specific gene (SVG) is a uniform expression value of 100 on the left half of the slide and 10 on the right side. Additionally, we generated 1,000 non-SVGs using spatial location permutations. As with all the experiments in the study, we applied BSP with default $D1 = 1$ and $D2 = 3$, and the adjusted p-values were then calculated using Benjamin-Hochberg corrections. The SVG was successfully detected, with an adjusted p-value < 0.0000 , with an acceptable false discovery (two false discoveries, one with an adjusted p-value of 0.005 and the other of 0.016). The detailed results can be found in **Extended Table 1**.

4. Hyper-parameter tuning. The spatial resolutions of different technologies vary dramatically across several magnitudes, from 50 nanometers (single-molecule fluorescence in situ hybridization such as MERFISH and SeqFISH+) to 100 microns (e.g., 10x Visium). The default settings (1 unit is defined as the distance that can capture one nearest neighbor, $D1$ is set to 1 unit, and $D2$ is set to 3 units) in different scenarios represent different physical distances and thus reflect different biological meanings. A systematic fine-tuning of these two key parameters for different types of data can help users to use this tool in their research.

Response: Thanks for the constructive suggestion. We have systematically explored fine-tuning the parameters in the BSP model. In the BSP model, $D1$ is normalized to a unit one by estimated density using Eq. (1) for 2D space and Eq. (2) for 3D volume. Once $D1$ is fixed to a unit of one, the only parameter that requires selection is $D2$, representing the granularity. The experiment details regarding this parameter selection have been outlined in **response to Critique #1(I)**.

Upon careful analysis, we observed that reasonably selecting $D2$ with the range of 2.0 to 5.0 exhibits insensitivity to the power of performance across three 3D patterns. Notably, we uniformly define $D2$ as 3 units, and this fixed parameter has proven to work well in all the simulations and case studies conducted across various SRT platforms and datasets in this study. This result suggests that BSP as a data-driven and non-parametric model is robust and effective.

5. The assumption of similar spatial resolutions at all three dimensions is true for STARmap, but not for

many other technologies such as MERFISH, for which the z-dimension resolution is several hundred folds lower than x- and y-dimensions. The performance evaluation of the tool in both simulated and real data (MERFISH/MERSCOPE, for example) will help users to understand whether this tool applies to such 3-D data.

Response: We appreciate the reviewer's comment on the potential variation in resolution between the z-dimension and the x- and y-dimensions. To address this concern, we designed an extended 3D simulation in which we decreased the resolution of the z-axis by multiplying the z-coordinates by ten ($z = 10, 20, \dots, 100$). This scenario allows us to replicate the 3D simulations with varying pattern sizes, signal strengths, and noise levels across three continuous 3D simulation patterns: curved stick (Pattern I), thin plate (Pattern II), and irregular lump (Pattern III), as described in Section 4.3.1. Considering the substantial differences between the z-coordinate and the x- or y- coordinate, we fixed the direction of the random walks in the z-axis. This strategy ensures that the sliced planes in our analysis can capture the simulated spatial pattern. The power analysis results indicate the BSP model can accurately identify SVGs in 3D SRT data, even when there are discrepancies in resolution between the inter-plane spatial resolution and the within-plane resolution. The description and revision of these findings are included in Lines 363-372 in Results Section 2.5 and Lines 687-698 in Methods Section 4.3.3:

Section 2.5

*In current 3D SRT, especially with data obtained from FISH-based sequencing techniques, it is common for the inter-plane spatial resolution (z-axis) to be considerably lower than the within-plane resolution (x- and y- axes)^{4,6}. To further assess the model's performance when the assumption of similar or equivalent spatial resolutions across all three dimensions is violated, we generated 3D simulations with varying scales of spatial resolutions between the within-plane (x- and y- axes) and inter-plane (z-axis) dimensions. Power analysis indicates that the BSP model accurately identifies SVGs in 3D SRT, even in scenarios with discrepancies between the inter-plane spatial and within-plane resolutions (**Figure 4F**). **Supplementary Figure 21** details a comprehensive power analysis on simulations with varying pattern sizes, signal strength, and noise levels.*

Section 4.3.3 3D simulation on FISH data with varying spatial resolutions in the z-axis

*In practical scenarios, the assumption of similar or equivalent spatial resolutions across all three dimensions may not hold, particularly in FISH-based sequencing techniques where the inter-plane spatial resolution (z-axis) is often significantly lower than the within-plane resolution (x- and y- axes). To systematically test the capability of BSP on 3D spaces, we increased the resolution of the z-axis by multiplying the z-coordinates by ten ($z = 10, 20, \dots, 100$) and replicated the simulation as described in Section 4.3.1. Considering the substantial differences between the z-coordinate and x- or y- coordinates, we fixed the direction of random walks in the z-axis to ensure that the sliced planes captured the simulated spatial pattern (**Supplementary Figure 42**). We varied the pattern sizes (small and moderate radius = 1.5 or 2.0), signal strengths (small and moderate, fold change = 2.0 or 2.5), and noise levels (small and moderate, $\tau=1$ or 2), respectively. The performance of the BSP model was compared with SPARK-X.*

Figure 4F) Power analysis for identifying SVGs in 3D simulations with inconsistent within-plane and inter-plane resolution.

6. Evaluating the performance of the 3D/BSP vs 2D/meta-analysis (especially the false discoveries) can be helpful. Could the authors perform simulated studies accordingly?

Response: Thank you for raising this important point. In the revision, we designed a series of 3D simulations stacking ten 2D slices on all continuous 3D patterns I, II, and III, as described in Section 4.3.1. These simulations were generated using moderate pattern size, signal strength, and noise level. To evaluate the performance of the BSP model, we performed power analysis and observed significant differences when applying BSP to the 3D data compared to the inference obtained from a meta-analysis based on the results from 2D slices. We included a new panel, **Figure 4C**, illustrating the power comparison between the BSP model on the 3D data and the meta-analysis on the 2D slices. A detailed description of these results is incorporated in Lines 350-355 in Section 2.5, under “BSP accurately and robustly identifies SVGs in 3D simulations”:

Section 2.5

*To demonstrate the advantages of utilizing 3D SRT, we compared SVGs identification performances between direct detection from 3D data and detection through meta-analysis, which combined individual analyses on each 2D slice using various combined probability tests. We performed simulations on all continuous patterns I, II, and III with moderate pattern size, signal strength, and noise level (**Figure 4C**). Our findings indicate that employing the BSP model on the 3D data yields superior power than the meta-analysis conducted on the results from 2D slices.*

Figure 4C) Power comparisons between 3D SRT and meta-analysis on 2D slices. Simulations were performed using a fixed moderate pattern size, moderate signal strength, and moderate noise level. The red line represents the p-value from BSP modeling the 3D SRT. Other colored lines depict p-values from various meta-analysis provided by the SciPy package, including Fisher, Pearson, Tippett, Stouffer, and Mudholkar, applied to BSP on 2D slices. Simulations using the 3D pattern I (curved stick), pattern II (thin plate), and pattern III (irregular lump) are shown in the left, middle, and right columns, respectively.

Minor issues:

1. A gene (MAN1A2) is identified as a spatially variant gene by meta-analysis but not when using the BSP (Fig 5 c). However, this is not obvious in Fig 5 c – it looks like this gene varies both in the x-y plane and in the z-direction. Could the authors to provide some statistical metrics to prove that those that are identified by meta-analysis using 2D stacks but not identified by BSP are false discoveries?

Response: Thank you for the insightful observation. We acknowledge the discrepancy between the meta-analysis approach, which combines individual 2D analyses under preset distribution hypotheses, and the BSP model, which directly models the stacked 2D slices as an intact 3D volume. We recognize that addressing this dimension-mismatch discrepancy requires significant theoretical developments in statistics, which are currently limited.

In our attempts to investigate this issue, we conducted simulations in **response to Critique 6**. However, as this is primarily a methodology-focused work, it may require additional research efforts and a separate publication to extensively investigate and interpret why certain genes, such as MAN1A2, may fail to be identified using the BSP approach.

To gain insights into the discrepancy, we also explored biological investigations. Due to the lack of benchmarks, we relied on annotations such as Gene Ontology and Pathway analysis from a biological perspective. Based on these annotations, it appears that the genes identified by the 3D BSP approach exhibit greater biological relevance to rheumatoid arthritis (RA) compared to the genes identified through meta-analysis.

We appreciate the reviewer's understanding of the challenges involved in addressing this discrepancy and the potential need for further research to explore the underlying reasons.

2. The maximum variance is used for normalization (Eq. 5). A potential issue is that outliers (which are common in spatial omics data) will dominate the scale of r_{D1D2} . Simulated studies on the sensitivity to

outliers (for example, spiking in some extremely large values) can help understand the robustness of the method. If the max-value-based normalization approach is not robust, the sigmoid transformation of log ratios (of the two variances), a commonly used robust transformation, can be an alternative for noisy data such as spatial omics data.

Response: Thanks for the suggestion. We provided the distribution quantile as a new user option '--quantileScaling' in BSP software to fit the distribution with upper 95% and lower 5%. Users can modify the quantiles in the BSP package to reconcile noisy data, such as spatial omics data.

3. For the beta distribution-based hypothesis test, which adjusted P-values are used? FDR?

Response: For the beta distribution, Bonferroni is used as the adjusted P-values. To enhance the flexibility, the newly introduced lognormal distribution does not have adjusted P-values where the users can make the corrections as needed.

4. It might be me – but the following links do not work: https://mailmissouri-581my.sharepoint.com/:f/g/personal/wangjue_umsystem_edu/EnjH6hbt1ptBjWBzoQhGfPoBhFVVy and <https://www.starmapresources.com/data>.

Response: We apologize for the problem. The URL is valid in our Word document, but it includes invalid characters in the converted PDF file. We updated the URLs with a mapping of a tiny URL: <https://tinyurl.com/2kaec9v4>. The website starmapresources.com worked when we conducted the analysis. Here is another URL in case it does not work: <https://www.wangxiaolab.org/data-portal-1>. We updated the corresponding URLs in Lines 730-733 in the Section of Data Availability.

Reviewer 3:

The manuscript by Juexin Wang et al. introduces the method BSP for identifications of spatially variable genes from spatial transcriptome data. BSP adapted the concept of spatial granularity to differentiate the SVGs and non-SVGs. The general methodology design is interesting and novel. The authors compared BSP with other existing tools to show superior performance of BSP. Analyses of published data with BSP revealed new SVGs and novel biological insights. This method could be potentially useful for the community. However, more extensive tests and more thorough discussion are needed to address critical issues and illustrate the pros and cons of the method. Following are my comments and questions.

Response: We thank the reviewer for all the constructive suggestions and critiques. We added several extended experiments and discussions. Please find our point-by-point responses below.

1. In general, I am convinced that the overall design of BSP should be suitable to capture some SVGs with clear patterns in large spatial domains. However, the readers do expect to see more extensive benchmarking of the method with simulated and real data, especially for the SVGs with different types of distribution patterns and in small domains. Following are just some special cases as examples that need to be tested.

a. For the small domains in tissue sections, for example, a tiny yet specially functional domain (let's say 1% of the tissue), would the SVGs be found to the right tail of the beta distribution in Fig. 1c? In other words, how efficient is BSP in identifying the SVGs of infrequent cell types or small tissue domains?

b. In another scenario, for the functional domains that are sparsely dispersed within a tissue section, would BSP efficiently identify the SVGs of these small and discrete domains?

Response: Thanks for the constructive suggestion. We designed additional simulations to evaluate the performance of BSP in identifying SVGs in **sparsely dispersed, small, and discrete domains**. As suggested by other reviewers, we designed a set of 3D simulation scenarios within the range of x- and y- coordinates from 0 to 30. The local discrete spatial patterns were constructed using solid spheres with a center-to-center distance of approximately 8 units. Sixteen center points were selected, where the z-coordinates were fixed at 5.5, and the x- and y- coordinates were generated using a sequence of numbers from 3 to 27 with an interval of 8. To introduce randomness to the spatial patterns, a uniformly distributed random variable ranging from -2 to 2 was added to each of the coordinates of the center points. The cells within the spheres were marked, and the expression values were assigned as described in Section 4.3.1. In this scenario, each sphere, representing a small domain, accounts for about 0.3% of the tissue, and the overall proportion of spiked cells is about 4%.

In this setting of simulations, SVGs were simulated using the irregular lump pattern described in Section 4.3.1. We generated 500 SVGs with discrete patterns influence locally and another 500 SVGs with continuous patterns influence globally, along with 9,000 non-SVGs as permuted genes without any spatial pattern. Three scenarios were considered to compare the effects of patterns' sizes, signal strengths, and noise levels, respectively. We adjusted radius values from 1.5 to 2.0 (small and moderate), expected expression fold changes from 2 to 2.5 (small and moderate), and noise levels from 2.0 to 3.0 (moderate and high). Ten replicates were generated to perform the power analysis.

The simulated discrete pattern is shown in **Figure 4D**. Power analysis indicates BSP model can accurately identify SVGs with these local discrete spatial patterns (**Figure 4E** and **Supplementary Figure 20**). The corresponding description and revisions are incorporated in Lines 356-362 in Results Section 2.5 and Lines 669-686 in Methods Section 4.3.2.

Section 2.5

*In addition to examining continuous patterns with global influence (depicted by the three 3D patterns in **Figure 4A**), we also evaluate the model performances on local discrete spatial patterns (**Figure 4D**). These patterns often manifest in isolated, small tissue domains in practical scenarios. Power analysis shows that the BSP model can accurately identify SVGs associated with these local discrete spatial patterns (**Figure 4E**). **Supplementary Figure 20** represents a comprehensive power analysis for simulations involving varying pattern sizes, signal strength, and noise levels.*

Section 4.3.2 3D simulation on FISH data with discrete spatial patterns

*To further evaluate the model's performance on data with locally influential discrete patterns (**Figure 4D**), we designed a set of 3D simulation scenarios within the ranges of x- and y-coordinates from 0 to 30. The local discrete spatial patterns were constructed using solid spheres with a center-to-center distance of 8 units. Specifically, sixteen center points were selected, with fixed z-coordinates of 5.5, and the x- and y- coordinates were generated from a sequence of numbers from 3 to 27 with an interval of 8. To introduce randomness into spatial patterns, we incorporated a uniformly distributed random variable ranging from -2 to 2 and added it to the coordinates of each center point. The cells within the spheres were marked, and expression values were assigned as previously described.*

In these simulations, SVGs were generated using the Irregular lump pattern described in 4.3.1. We created 500 SVGs with locally influential discrete patterns and an additional 500 SVGs with globally influential continuous patterns, along with 9,000 non-SVGs as permuted genes without any spatial pattern. We considered three scenarios to compare the effects of pattern sizes, signal strengths, and noise levels, respectively. We adjusted radius values from 1.5 to 2.0 (small and moderate), expected expression fold changes from 2 to 2.5 (small and moderate), and noise levels from 2.0 to 3.0 (moderate and high) as previously described. To ensure robustness, ten replicates were generated to perform the power analysis.

2. The author only compared BSP with just a couple of the existing methods. This is far from enough. Other methods should be included for the comparisons with simulated and real data.

Response:

In response to the reviewer's suggestion, we added nnSVG as an additional method for comparison in the revised manuscript. The nnSVG method, as described in the paper "nnSVG for the scalable identification of spatially variable genes using nearest-neighbor Gaussian processes" (Weber, Lukas M., et al. Nature Communications 14.1 (2023): 4059), has been evaluated against several methods that we also included in the BSP paper, such as SpatialDE, SPARK, SPARK-X, Moran's I, and another baseline method HVGs we did not include in our analysis (Amezquita, R. A. et al. Orchestrating single-cell analysis with Bioconductor. Nat. Methods 17, 137–145 (2019)). According to their analysis, nnSVG showed better or comparable performance to these competitive methods. Furthermore, it is worth noting that the

nnSVG method was recently published in **Nature Communications** on **July 10th, 2023**. Given its state-of-the-art performance and its recent publication, we believe that nnSVG can be considered a representative method in the field nowadays. Therefore, including nnSVG in our comparison will offer a comprehensive assessment of BSP's performance against one of the latest and most competitive methods in SRT analysis.

In the revision, we included nnSVG in all the analyses along with other competitive methods such as spatialDE, SPARK, SPARK-X, and Moran's I. These methods have been used in 2D simulations with varying pattern sizes, signal strength, and noises, as well as in real data studies. Our comprehensive evaluation of 2D simulations and benchmarks using mouse olfactory bulb data and human breast cancer data shows that BSP is accurate and fast compared to the most competitive methods. On 3D simulations, BSP demonstrates superior accuracy compared to the competitive methods.

(A) In performance comparisons on simulations, **Figure 2B** in the following shows the power comparison among different methods using moderate pattern size, noise level, and signal strength for the spatial expression patterns I, II, and III (left to right) in the simulations. To provide a more comprehensive power analysis, we have included additional comparisons considering different signal-to-noise ratios (weak, moderate, and strong) with a moderate noise level in the newly added **Supplementary Figure 1**. Moreover, we have examined different noise levels (low, moderate, high) with a moderate signal-to-noise ratio, as shown in the newly added **Supplementary Figure 2**.

We also conducted computational time and memory usage comparisons with other competitive methods on an Ubuntu 16.04.4 LTS workstation with Intel(R) Xeon(R) W-2125 CPU @ 4.00GHz and 32 GB memory in the newly added **Supplementary Table 1**. **Figure 2C** in the following shows computational time costs with an increasing number from 2,000 genes to 20,000 genes and fixed 2,000 spots. **Figure 2D** shows the computational time costs with an increasing number from 1,000 spots to 10,000 spots and fixed 10,000 genes. The corresponding memory usage is shown in the newly added **Supplementary Figure 3**.

The updated analysis and benchmarking have been included in the revised manuscript in Lines 152-192 in the Results Section 2.2 and corresponding Figure 2, Supplementary Figures 1-3, Supplementary Table 1.

(B) We also tested the performance of nnSVG on real data benchmarks on mouse olfactory bulb. Unfortunately, the R package of nnSVG continuously throws errors on the human breast cancer dataset. We raised the issue on the nnSVG Github page <https://github.com/lmweber/nnSVG/issues/16>. The updated analysis and benchmarking have been included in the revised manuscript Lines 201-207 in Results Section 2.3, corresponding to Figure 3A and Supplementary Figure 4.

3. The authors claimed that BSP is non-parametric. One would anticipate that different patch sizes, for both the small and large patches, should lead to different sensitivity and specificity of BSP. This needs to be tested, again, with simulated and real data.

Response: Thanks for the constructive suggestion. We have conducted a systematic exploration of fine-tuning the parameters in the BSP model per the review's recommendation. In the BSP model, D1 is normalized to a unit one by estimated density using Eq. (1) for 2D space and Eq. (2) for 3D volume. Once

D1 is fixed to a unit of one, the only parameter that requires selection is D2, representing the granularity. The exploration has been utilized in **(A)** the added simulations using three 3D spatial patterns, and **(B)** the real data benchmarks of mouse olfactory bulb and human breast cancer.

(A) In the 3D simulations in all three 3D spatial patterns, including curved stick (Pattern I), thin plate (Pattern II), and irregular lump (Pattern III) described in Section 4.3.1, the power analysis results are presented in the newly added **Supplementary Figure 22**. After checking the impact of various D2 radius values with 2.0, 2.5, 3.0, 3.5, 4.0, 5.0, and 10.0, we observed that the BSP method exhibits insensitivity to the parameter selection of D2 within the range of 2 to 5. It is also noteworthy that the statistic power decreases when an excessively large D2 is used (e.g., D2=10). Based on experimental results across all the simulations and case studies conducted on different SRT platforms and datasets in this work, we uniformly define D2 as 3, which has proven to yield satisfactory results.

(B) In the real data benchmarks of mouse olfactory bulb and human breast cancer, we employed a gradient of D2 values ranging from 2.0 to 10.0 to evaluate the performance of BSP in identifying marker genes as reported in their original studies. The mouse olfactory bulb dataset consists of 10 marker genes, and the human breast cancer dataset includes 14 marker genes (as described in Section 2.3). From the results presented in the newly added **Supplementary Table 36**, we observed that the performances of BSP are basically stable with ranges of D2 selection in the real data benchmarks. For the mouse olfactory bulb study, BSP consistently identifies more than 8 marker genes when D2 values range from 2 to 8. Similarly, in human breast cancer study, BSP successfully identifies 13 marker genes when D2 values range from 2 to 4.

These revisions, findings, and descriptions have been incorporated into Lines 373-379 in Section 2.5 and Lines 572-575 in Section 4.1.1

Section 2.5

*Specially, BSP consistently identifies SVGs regardless of selecting different radius values on the scale of the big patch (D_2) in contrast to the small patch (D_1). Following co-clustering strategies¹⁹, we set the small patch as the reference unit with a fixed value one, power analysis reveals slight differences when D_2 ranges from 2.0 to 5.0. The same trends can be observed in all three 3D continuous patterns presented in **Supplementary Figure 22**. By making reasonable choices ($D_2 \leq 5.0$), BSP exhibits insensitivity to parameter selection, enabling its application across a wide range of SRT platforms and datasets.*

Section 4.1.1

*...More comprehensive exploration shows the BSP performance is insensitive in selecting D_2 on scenarios of simulations (**Supplementary Figure 22**) and on real data of mouse olfactory bulb and human breast cancer studies (**Supplementary Table 36**).*

Upon careful analysis, we observed that reasonably selecting D2 with the range of 2.0 to 5.0 exhibits insensitivity to the power of performance across three 3D patterns. Notably, we uniformly define D2 as 3 units, and this fixed parameter has proven to work well in all the simulations and case studies conducted across various SRT platforms and datasets in this study. This result suggests that BSP as a data-driven and non-parametric model is robust and effective.

4. Pattern II in Fig. 2B clearly shows that BSP does not perform as well as SPARK or SpatialDE. The

statement in from line161 to 163 is simply not true. This is related to my first comment. It is very likely that BSP works well with some patterns of SVGs but not with other special patterns. The authors should explicitly show the scenarios in which BSP may not work well via more comprehensive tests with real and simulated data.

Response: We double-checked the results and revised the statement in Lines 170-172 as *“On the second pattern, although the weak and moderate signal strength limits the performance, BSP exhibited better power when the signal strength was high (4-fold) (Supplementary Figure 2).”*. This statement relates to the highlighted red subfigure in **Supplementary Figure 1**.

To comprehensively test the simulation data, we performed 2D simulations with different signal strength and noise level combinations in three spatial patterns in **Supplementary Figure 1** and **Supplementary Figure 2**. We acknowledge that BSP may not work well for a large portion of combinations of signal strengths and noise levels in Pattern II. Given the complexity of SVGs, it is unlikely that one tool outperforms all others. The intent of our development is not to replace other tools, but to add a highly competitive tool to improve the state of the art.

In real data, we want to mention that these three 2D spatial patterns are clustered on genes identified by SPARK. In practice, BSP achieved great performance on the mouse olfactory bulb data benchmark. Lines 201-207 in Section 2.3 states that *“...BSP detected 9 of 10 marker genes from the original study, while SpatialDE detected 3, SPARK detected 8, nnSVG detected 6, and SPARK-X detected 0. The only marker gene BSP missed was Sv2b with a marginal p-values of 0.0518. This missed marker gene (Supplementary Figure 5) may not has expression variances confined to many isolated, relatively small regions, which could result in the same variances in both big and small patches...”*.

To compare the statistical power, we examined different signal-to-noise ratios (FC = 3,4,5) with a moderate noise level ($\tau = 0.5$ as defined in SPARK) in **Supplementary Figure 1**. Additionally, we compared the methods' performance under different noise levels ($\tau = 0.2,0.5,0.8$) with a moderate signal-to-noise ratio (FC = 4), as shown in **Supplementary Figure 2**.

Supplementary Figure 1. Power comparison of different methods with varying signal strengths in 2D simulations. Power charts show the averaged true positive rates (y-axis) across ten replicates against the false discovery rates (x-axis) for the detected SVGs using each method. In these nine power charts, simulations with weak, moderate, and high signal strengths are shown in the left, middle, and right columns, respectively. Simulations using the spatial expression patterns I, II, and III are placed in the top, middle, and bottom rows, respectively. All simulated datasets were generated using a fixed moderate noise level.

Supplementary Figure 2: Power comparison of different methods with varying noise levels in 2D simulations. Power charts show the averaged true positive rates (y-axis) across ten replicates against the false discovery rates (x-axis) for the detected SVGs using each method. These nine power charts show simulations with low, moderate, and high noise levels in the left, middle, and right columns, respectively. Simulations using the spatial expression patterns I, II, and III are placed in the top, middle, and bottom rows, respectively. All simulated datasets were generated using fixed moderate signal strength.

Supplementary Figure 5. Missed marker gene *Sv2b* by BSP in mouse olfactory bulb study. Colors indicate gene expression levels. The expression values were log-transformed, and those greater than 1.0 were normalized to 1.0.

5. Data drop-out is a major issue of current spatial transcriptome data. The robustness of BSP to forced drop-outs needs to be tested.

Response: Thanks for the suggestion. We have conducted a series of new simulations to evaluate BSP's performance of BSP with different ratios of drop-out effects. The power analysis results, as presented in the newly added **Supplementary Figure 19**, demonstrate the robustness of BSP in the presence of reasonable dropout rates across all scenarios.

We added the corresponding paragraph Lines 344-349 in Section 2.5:

*We further investigated the capability of BSP on SRT in the presence of dropout issues⁴⁴, where only a subset of the transcriptome is captured by sequencing due to technical limitations. Dropout effects were simulated by randomly assigning zero-values to a certain proportion (10%, 20%, and 30%) of cells in continuous patterns I, II, and III with moderate pattern size, signal strength, and noise level. The power analysis (**Supplementary Figure 19**) indicates the robustness of BSP in handling SRT with a reasonable dropout rate.*

6. Presentations of the SVGs identified by BSP with real data are descriptive. Discussion of the biological relevance of the SVGs was simply based on GO analyses. Such results are barely informative. Most of the GO terms were too general to derive any meaningful biological insights. The authors need to do more thorough and deeper analyses of the SVGs, which should be classified into groups based on their distribution patterns.

Response: Thank you for this important critique. Your suggested analysis has enhanced our study significantly.

A) We have conducted additional hierarchical clustering to identify two distinct spatial patterns of SVGs in kidney studies in Section 2.3. We have worked with a nephron-biologist (Dr. Michael Eadon) and a clinical nephrologist (Dr. Krzysztof Kiryluk) to help interpret these patterns. The two most common causes of tubulointerstitial AKI are 1) acute tubular necrosis which is caused by ischemia or nephrotoxicity, and 2) acute interstitial nephritis, which is caused by an immune reaction. The first pattern is enriched for oxidative phosphorylation and aerobic respiration genes, which indicate acute tubular necrosis. The second pattern includes a variety of immune-mediated pathways, consistent with interstitial nephritis. Since tubular necrosis is treated with conservative management and interstitial nephritis is treated with immunosuppression, the ability to distinguish these molecular contributions on a kidney biopsy sample holds real clinical utility.

In addition, AKI is characterized by stages of injury which include a transitional stage of translational shutdown, followed by recovery. Both pathways were identified in separate localized regions of the kidney biopsy samples and may provide prognostic significance as to the potential for recovery. Thus, the data suggest that BSP can help distinguish the underlying cause of AKI as well as the localized temporal stage of AKI within the tissue.

Clinically, the patient donated his kidney for SRT sequencing, and the data are available in the kidney atlas by Kidney Precision Medicine Project (KPMP). The authors used this data to demonstrate the capacity of BSP as a case study. Here are the notes from Dr. Krzysztof Kiryluk, who biopsied the patient at Columbia University.

“It is a 71 y/o M with polysubstance abuse who used intranasal heroin, fell out of bed while high, and was down for an unspecified period of time. Developed bad rhabdomyolysis and severe (stage 3) AKI with CK level of 95K. He was biopsied 2 weeks after initial presentation, already in the recovery stage when his mental status and renal function were improving (serum creatinine 3.46 mg/dL at the time of biopsy, decreased from the peak of 11.46 mg/dL).

Renal biopsy showed the following:

1. Tubular degenerative and regenerative changes, diffuse, moderate, consistent with acute tubular injury, with myoglobin casts (rhabdomyolysis-associated/clinical).
2. Tubular atrophy and interstitial fibrosis, mild.
3. Arterio- and arteriolosclerosis, mild.
4. Mesangial C3 deposits, suggestive of resolving/resolved phase of infection-related glomerulonephritis.”

We added the corresponding revisions and descriptions in Lines 217-260 in Section 2.3:

We next extended the application of BSP to the study of Acute Kidney Injury (AKI)²⁷. We ran BSP on a human kidney biopsy sample with 10X Visium data collected and processed by the Kidney Precision Medicine Project²⁸. The biopsy was performed on a 71-year-old Hispanic man two weeks after his initial presentation with severe (stage 3) AKI in the setting of rhabdomyolysis due to a heroin overdose. The biopsy showed acute tubular injury with myoglobin casts (rhabdomyolysis-associated) and diffuse tubular degenerative and regenerative changes, mild interstitial fibrosis, and superimposed C3 mesangial deposits suggestive of resolving infection-related glomerulonephritis.

*BSP identified 285 SVGs (p -value <0.05) consisting of 317 spots and 14,988 genes. Annotated by clusterProfiler²⁹, the results were supported by gene ontology (GO) enrichment analysis (**Figure 3C**), including relevant enrichments in humoral immune response (q -value $1.09e-11$), and humoral immune response mediated by circulating immunoglobulin (q -value $1.63e-10$). Reactome enrichment analysis³⁰ identified eukaryotic translation elongation (q -value $2.77e-13$) and influenza infection (q -value $5.59e-09$), both indicative of the translational shutdown phase of AKI³¹ (**Supplementary Figure 7**). As innate and adaptive immune responses spatially and temporally correspond with damage to renal tubular cells and recovery from AKI^{32,33}, these results are consistent with disease enrichment analysis³⁴. Notably, the disease enrichment analysis revealed highly significant terms related to urinary system disease (q -value $9.30e-11$) and kidney disease (q -value $1.36e-10$). **Supplementary Table 6** lists all the SVG results obtained by BSP. **Supplementary Table 7** details the results from GO enrichment analysis, **Supplementary Table 8** describes the results from Reactome, and **Supplementary Table 9** details the results from GO enrichment analysis.*

*To further investigate the functionalities of SVGs, hierarchical clustering identified two spatial patterns from the kidney sample in **Figure 3D**. Pattern 1 includes 235 genes. GO enrichment analysis indicated genes with this expression pattern participated in aerobic respiration (q -value $1.04e-09$) and oxidative phosphorylation (q -value $4.69e-09$), consistent with the main pathologic diagnosis of acute tubular necrosis. Pathways of early recovery were also enriched, including kidney development (q -value $8.56e-07$) and metanephric nephron epithelium development (q -value $2.30e-06$), which included genes like PAX8. Pattern 2 included 50 genes. GO enrichment analysis indicated genes with this expression pattern were related to humoral immune response (q -value $5.15e-05$) and tissue homeostasis (q -value $5.15e-05$). Several immune responses were activated (q -value $3.51e-04$), including the B cell receptor signaling pathway (q -value $4.72e-07$). The immune-related pathways are potentially consistent with an inflammatory response to acute tubular injury and potentially resolving infection-related glomerulonephritis. Acute tubular necrosis is characterized by stages of injury, including a transitional stage of translational shutdown, followed by recovery. Both of these pathways were identified in separate localized regions of the kidney biopsy samples and may provide prognostic significance as to the potential for recovery. Thus, our results demonstrate that BSP is able to identify relevant transcripts corresponding to a specific AKI subtype and provide information on the severity (active necrosis and inflammation) and the temporal stage of AKI (evidence of recovery in this case). **Supplementary Tables 10-13** and **Supplementary Figures 8-11** detail the results from the GO enrichment and KEGG pathway analyses for both patterns. By demonstrating BSP's utility in kidney research, our study highlights the potential for BSP to differentiate the underlying complex disease subtypes in diverse tissue samples.*

Figure 3D) Hierarchical clustering identified two spatial patterns of SVGs. The left branch (Orange) is Pattern 1, represented by gene *MT1G* in the kidney sample using 10X Visium. The right branch (Green) is Pattern 2, represented by gene *IGKC* in the kidney sample using 10X Visium.

Supplementary Figure 8: GO enrichment analysis on SVGs of Pattern 1 in AKI study using 10X Visium.

Supplementary Figure 9: Pathway enrichment analysis on SVGs of Pattern 1 in AKI study using 10X Visium.

Supplementary Figure 10: GO enrichment analysis on SVGs of Pattern 2 in AKI study using 10X Visium.

Supplementary Figure 11: Pathway enrichment analysis on SVGs of Pattern 2 in AKI study using 10X Visium.

B) We also have conducted additional hierarchical clustering to identify four distinct spatial patterns of SVGs in RA research using 3D SRT in Section 2.6. We added the corresponding paragraph in Lines 456-472 in Section 2.6:

To explore the functionalities of SVGs further, we performed hierarchical clustering on the RA1 sample, resulting in the identification of four distinguishable spatial patterns (Supplementary Figure 25). Pattern 1 consisted of 91 SVGs and exhibited enrichment in ribosome genes associated with biogenesis pathways (q-value 3.10e-16). This pattern suggests a general biosynthetic functionality supporting RA tissue across the highlighted

regions⁵⁰. Pattern 2 comprised 774 SVGs, showing enrichment in lymphocyte proliferation (q-value 2.22e-07) and activation of the immune response (q-value 1.69e-06), indicating an immune activation region. Notably, the presence of CXCL13 and MS4A1 in this pattern suggests the onset of inflammation in RA⁵¹. Pattern 3 consisted of 120 SVGs and enrichment in apoptosis activity (q-value 7.31e-06) and complement activation process in the pathogenesis of RA (q-value 1.96e-05), supported by genes such as C1S, C1R, C1QC, S100A8, and S100A9⁵². Similarly, Pattern 4 included 233 SVGs and showed enrichment in apoptosis (q-value 4.33e-05) and regulation of leukocyte migration (q-value 1.11e-03), indicating the formation of chronic inflammation and autoimmunity in RA development through recruiting leukocytes⁵³. **Supplementary Tables 26-33** and **Supplementary Figures 26-33** provide detailed results from the GO enrichment and KEGG pathway analyses for all four patterns. This analysis underscores the opportunities afforded by BSP analysis on intact 3D volumes in identifying SVGs compared to potentially biased 2D analysis.

Supplementary Figure 25: Hierarchical clustering identified four spatial patterns of SVGs of the RA1 sample using 3D SRT. From left to right: Pattern 4, represented by gene ACP5; Pattern 2, represented by gene ABCC3; Pattern 3, represented by gene A2M; Pattern 1, represented by gene ACTB.

7. Overall, in my mind, we are not in an urgent need for another tool for identifying the SVGs. There are already many tools available. New methods like BSP would be more appreciated if they can provide further and deeper information of the SVGs, for example, like I mentioned above, classifications of the SVGs based on their distribution patterns, inference of cell types and tissue domains related to the SVGs, functional associations between different domains of the tissues as indicated by the SVG distributions, etc.

Response: We greatly appreciate many of your suggestions that help us provide further and deeper information of the SVGs. For example, the additional analyses and interpretation in **response to critique #6** offer a deeper understanding and utility for the SVGs derived from BSP, which allow us to interpret the underlying cause of disease as well as the stage of recovery.

Reviewer #1 (Remarks to the Author):

First of all, I really appreciate the authors' effort in reviewing the manuscript according to the questions from the reviewers.

I just have a couple of questions left.

- While you have compared the goodness of fit between the beta and lognormal distributions, I would also suggest considering the utilization of empirical quantiles. This approach introduces a completely nonparametric dimension to the method presented in this manuscript. I encourage conducting a few experiments using this approach, and based on the results, consider incorporating the use of empirical quantiles as a third option within your method, alongside the two parametric ones.

- Have the authors identified any specific frameworks or scenarios where their proposed method has encountered criticism or limitations? I believe that acknowledging any criticisms does not weaken the manuscript but, rather, aids users in selecting the appropriate method for data analysis. If such criticisms exist, I encourage the authors to address them in the final discussion. This addition will not diminish my positive opinion of this work.

Reviewer #2 (Remarks to the Author):

The authors have addressed most of my comments decently and provided exciting new evidence and insights into the nature of the model and the underlying reasons why this model performs well. The reviewer appreciates very much the wonderful responses from the authors.

Meanwhile, there are minor issues that might potentially further improve this manuscript. The reviewer would recommend minor revisions.

Comment 1 (I) (A) and (B): Just wondering whether the authors would like to discuss the relation between different application scenarios and the optimal gradient. The authors have shown that, for identifying marker genes, a ratio between 2 – 8 in one case and 2 – 4 in another case achieves the most consistent results with the literature. It would be helpful for users and the audience if the authors could provide some biological insights and practical guidance on how to choose this ratio.

Comment 2 (II) (A): The explanation of why the ratio works, "the velocity of changes in the variances of local means", is inspiring. Would the authors consider including this point in the main text?

Comment 3. Maybe providing some discussion/thoughts on why the model still performs well when the radiuses are significantly smaller than the spatial pattern?

Comment 3: Scalability could be an issue, as new spatial technologies quickly reach and exceed the 1,000-by-1,000 resolution. For example, the Stereo-seq technology now reaches a spatial resolution of 500nm (a center-to-center distance between spots) on slides of size 65mm-by-65mm or even bigger (PMID: 35512702). The size of a data-capturing unit is 10mm-by-10mm. This gives 20,000-by-20,000 spots from each unit and an overall 130,000-by-130,000 spot array of the whole slide. Maybe some discussions on potential solutions for scalability will help. It is not necessary to address this issue in this work.

Minor issues: 1: Sorry for the unclear expression – no need to biologically prove or investigate this issue since this is a methodology work. The reviewer asked whether the authors could provide statistical metrics to measure the spatial variance of genes and thus to quantitatively demonstrate that the apparent spatial patterns of genes such as MAN1A2 are actually due to random noises. There are well-established statistical tools in spatial statistics, especially in spatial heterogeneity analysis, such as q-statistics, HTA index, ht-index, etc. Would it be practical to provide a quantitative comparison, using a spatial statistical index, between the genes of interest (such as MAN1A2) with randomly generated spatial data (as a true negative control) on the same slide and show that MAN1A2 is actually not statistically significantly different from such random noises?

Reviewer #3 (Remarks to the Author):

The authors have performed a series of new and more insightful analyses in their revision. Most of my previous critiques have been addressed. I have no further comments.

Reviewer #1

First of all, I really appreciate the authors' effort in reviewing the manuscript according to the questions from the reviewers. I just have a couple of questions left.

Comment 1: While you have compared the goodness of fit between the beta and lognormal distributions, I would also suggest considering the utilization of empirical quantiles. This approach introduces a completely nonparametric dimension to the method presented in this manuscript. I encourage conducting a few experiments using this approach, and based on the results, consider incorporating the use of empirical quantiles as a third option within your method, alongside the two parametric ones.

Response: We appreciate the valuable feedback. Based on the ratio of the variance of local mean in each gene, using empirical quantiles in conjunction with the ranking of the testing scores offers the potential to be a nonparametric approach in the current algorithm. This flexibility allows users to define SVGs as the top-ranking genes by specifying an arbitrary quantile according to their domain knowledge. We added a feature in our software for users to output top n% genes.

As suggested, we conducted experiments on both the ST Mouse Olfactory Bulb dataset and the ST Human Breast Cancer dataset using empirical quantiles. We considered different quantiles, including the top 1%, 2%, 3%, 4%, 5%, 10%, 15%, and 20% ranking genes as SVGs in both datasets. The results revealed the challenge of finding a consistent quantile across different studies. For instance, at the 5% quantile, the Mouse Olfactory Bulb study identified half of the 10 marker genes from the original literature (5 out of 10). In contrast, the Human Breast Cancer study required a 3% quantile to identify half of the 14 marker genes (7 out of 14). It was not until the 15% quantile that the Mouse Olfactory Bulb study identified all the marker genes, while the Human Breast Cancer study needed a 20% quantile to do the same (Extended Table 1). In contrast, using gene rankings along with their corresponding p -values based on the lognormal distribution (Extended Table 2), extracted from Supplementary Table 1 and Supplementary Table 2, provides a much more robust confidence assessment.

As shown in the above experiments, the suitability of an empirical quantile can vary significantly across different biological contexts and sequencing technologies, making it challenging to establish a universal quantile applicable in most scenarios. Consequently, we employed statistical distributions, such as the beta and lognormal distributions, to model the distribution of the testing scores. We assume these distributions provide a concise representation of the underlying nature of SVGs within the spatial samples. By assigning p -values to each gene based on the fitted distribution, we can uniformly use a fixed cutoff, typically set as 0.05, across all the simulations and all experiments in our study. As utilized in previous methods such as SPARK and SPARK-X, this distribution-fitted approach is usually robust to varying levels of noise and widely accepted within the field.

In conclusion, we provide this handy option for users to explore the top-quantile genes in the BSP software using empirical quantiles, while we do not recommend using empirical quantiles for confidence assessment of SVG identification.

Mouse Olfactory Bulb (Total 12,602 genes)		Human Breast Cancer (Total 8,434 genes)	
Quantile	# of Identified SVGs from 10 marker genes within quantile	Quantile	# of Identified SVGs from 14 marker genes within quantile
1% (126)	0	1% (84)	6
2% (252)	2	2% (169)	6
3% (378)	3	3% (253)	7
4% (504)	3	4% (337)	8
5% (630)	5	5% (422)	9
10% (1260)	8	10% (843)	11
15% (1890)	10	15% (1265)	13
20% (2520)	10	20% (1691)	14

Extended Table 1. Empirical quantiles on the real datasets ST Mouse Olfactory Bulb and ST Human Breast Cancer. Numbers in the parentheses is the absolute gene number with the percentage in each study.

Mouse Olfactory Bulb (10 Marker Genes)			Human Breast Cancer (14 Marker Genes)		
Ranking	Gene Name	p-value	Ranking	Gene Name	p-value
148	Doc2g	0.00019684	1	POSTN	0
182	Cdhr1	0.00031357	5	FN1	0
342	Slc17a7	0.00141615	19	SPARC	6.74E-13
520	Uchl1	0.00369997	25	VIM	6.72E-12
584	Reln	0.00461596	39	DCN	2.20E-10
935	Plcx2	0.01444757	40	IGFBP5	2.27E-10
1025	Rcan2	0.01744832	175	MMP14	6.11E-06
1243	Shisa3	0.0251025	321	SCGB2A2	0.00018818
1370	Nmb	0.03078843	389	KRT17	0.00040459
1794	Sv2b	0.05182136	440	MUCL1	0.00063843
			544	GAS6	0.00143679
			918	AREG	0.01345101
			1259	PEG10	0.03880608
			1684	PIP	0.08499101

Extended Table 2. The rankings and corresponding p -values of the marker genes on the real datasets ST Mouse Olfactory Bulb and ST Human Breast Cancer.

Comment 2: Have the authors identified any specific frameworks or scenarios where their proposed method has encountered criticism or limitations? I believe that acknowledging any criticisms does not weaken the manuscript but, rather, aids users in selecting the appropriate method for data analysis. If such criticisms exist, I encourage the authors to address them in the final discussion. This addition will not diminish my positive opinion of this work.

Response: Thanks for the constructive suggestions. We extended the discussions on encountered criticisms and limitations as follows in lines 541-557:

Although BSP has shown notable advancements in quantitatively measuring spatial patterns using the lognormal distribution to fit the distribution of test scores of all the genes, some limitations still need to be addressed. Through meticulous examination of histograms on permuted data vs. density of distribution and corresponding Q-Q plots (Supplementary Figures

39-40), and goodness-of-fit test with Cramer-von Mises criterion (Supplementary Tables 34-35), it becomes evident that the lognormal distribution offers a more suitable fit than the beta distribution for studies involving 2D simulation data, mouse olfactory data, and human breast cancer data. However, alternative statistical distributions or non-statistical ranking measurements could be explored to further improve the fitting of the distribution of ratios between variances of the averaged expression in the paired big-small patch. In the practical usage where spatial patterns exhibit alternating high- and low-expressed cells within a confined area, e.g., a pattern of thin curved stick in Figure 4A, it is essential to exercise caution regarding the choice of patch radius. Using an excessively large patch radius can potentially result in a reduction in statistical power, as demonstrated in Supplementary Figure 22. Furthermore, BSP compromises the performance and computational resources in SRT studies. Although it performs better on the benchmarks, BSP consumes more time and memory than SPARK-X on large-scale datasets.

Reviewer #2

Comment 1 (I) (A) and (B): Just wondering whether the authors would like to discuss the relation between different application scenarios and the optimal gradient. The authors have shown that, for identifying marker genes, a ratio between 2 – 8 in one case and 2 – 4 in another case achieves the most consistent results with the literature. It would be helpful for users and the audience if the authors could provide some biological insights and practical guidance on how to choose this ratio.

Comment 2 (II) (A): The explanation of why the ratio works, “the velocity of changes in the variances of local means”, is inspiring. Would the authors consider including this point in the main text?

Comment 3. Maybe providing some discussion/thoughts on why the model still performs well when the radiuses are significantly smaller than the spatial pattern?

Response: We extend our gratitude to the reviewer for the valuable insights and comments. To address Comments 1, 2, and 3, we would like to provide a detailed explanation of the underlying mechanism of the BSP framework:

Discussion:
Lines 513-526:

Specifically, BSP operates by assessing how rapidly the variances of local means change as the radius of the patch is adjusted. The primary source of these variance fluctuations arises from neighboring regions that exhibit distinct expression levels, where the local means within such regions change more gradually as the patch radius varies than between neighboring cells or spots. Consequently, the velocity of changes in the variances of local means for a gene with global spatial patterns is comparatively slower than for the genes that lack discernible spatial patterns. This behavior becomes notably prominent when the patch radius is chosen within a reasonable range. Although the patch radius is significantly smaller than the spatial pattern, the patches are averaged over all positions on the whole spatial transcriptomic space and hence the model can capture the global patterns of SVGs. Therefore, we recommend using the default value of 3.0 for the radius of large patches in most situations, especially when the sizes of

spatial patterns remain unclear. Users may adjust the value of D2 within the range of 2.0 to 5.0, particularly when dealing with a known, large-scale spatial pattern.

Lines 550-554:

In cases where spatial patterns exhibit alternating high- and low-expressed cells within a confined area, e.g., a pattern of thin curved stick in Figure 4A, it is essential to exercise caution regarding the choice of patch radius. Using an excessively large patch radius can potentially result in a reduction in statistical power, as demonstrated in Supplementary Figure 22.

We added the paragraphs in Lines 513-526 and 550-554 in the Discussion Section.

This addressed **Comment 2** by adding the explanation of why the ratio works in the main text, as well as **Comment 3**. For practical guidance on how to choose the ratio on **Comment 1**, we recommend using the default value of 3.0 in most situations, especially when the sizes of patterns remain unclear. Users have the option to adjust the value of D2 within the range of 2.0 to 5.0, particularly when dealing with a known, large-scale spatial pattern.

Comment 3: Scalability could be an issue, as new spatial technologies quickly reach and exceed the 1,000-by-1,000 resolution. For example, the Stereo-seq technology now reaches a spatial resolution of 500nm (a center-to-center distance between spots) on slides of size 65mm-by-65mm or even bigger (PMID: 35512702). The size of a data-capturing unit is 10mm-by-10mm. This gives 20,000-by-20,000 spots from each unit and an overall 130,000-by-130,000 spot array of the whole slide. Maybe some discussions on potential solutions for scalability will help. It is not necessary to address this issue in this work.

Response: We appreciate the reviewer's valuable insight into the rapid development of high-resolution spatial transcriptomic sequencing techniques. We are exploring the incorporation of sparse matrices to accelerate the computational processes on high-resolution spatial transcriptomics data. This enhancement will be integrated into the following versions. We added the corresponding sentences in Lines 561-563 in the Discussion Section.

Minor issues: 1: Sorry for the unclear expression – no need to biologically prove or investigate this issue since this is a methodology work. The reviewer asked whether the authors could provide statistical metrics to measure the spatial variance of genes and thus to quantitatively demonstrate that the apparent spatial patterns of genes such as MAN1A2 are actually due to random noises. There are well-established statistical tools in spatial statistics, especially in spatial heterogeneity analysis, such as q-statistics, HTA index, ht-index, etc. Would it be practical to provide a quantitative comparison, using a spatial statistical index, between the genes of interest (such as MAN1A2) with randomly generated spatial data (as a true negative control) on the same slide and show that MAN1A2 is actually not statistically significantly different from such random noises?

Response: We appreciate the reviewer's suggestion regarding the need for quantitative validation. For the genes exclusively identified in the 2D meta-analysis, we assessed spatial stratified heterogeneity (SSH) as a form of quantitative validation. We calculated SSH and the corresponding *p*-values with q-statistics for both x- and y-coordinates, assuming a null

hypothesis of no stratified heterogeneity [ref: Wang, Jin-Feng, Tong-Lin Zhang, and Bo-Jie Fu. "A measure of spatial stratified heterogeneity." *Ecological indicators* 67 (2016): 250-256.]. As an illustrative example, we considered the gene MAN1A2 (only detected in 2D meta-analysis), which yielded p -values of 0.2163 for x-coordinates and 0.3189 for y-coordinates. These p -values indicate the absence of statistically significant stratified heterogeneity for MAN1A2. In contrast, the p -values of gene SEMA4D (only detected in 3D analysis) are 0.0019 and 0.0004 for x- and y-coordinates, respectively, signifying statistically significant stratified heterogeneity.

It is notable that while various statistics exist for measuring spatial autocorrelations (e.g., Moran's I) or spatial heterogeneity (e.g., q -statistics), there is no universally accepted computational gold standard to determine whether a gene qualifies as an SVG. In our study, we thoroughly reviewed the performance of alternative test statistics, such as Moran's I, and found that their performance was suboptimal, as depicted in Figure 2b. Additionally, these methods often incurred substantial computational time and memory requirements, as illustrated in Figure 2c. We added the corresponding sentences in Lines 445-455 in the Results Section.

Reviewer #3 (Remarks to the Author):

The authors have performed a series of new and more insightful analyses in their revision. Most of my previous critiques have been addressed. I have no further comments.

Response: Thanks for the confirmation.

Reviewer #1 provided comments to the editorial team supportive of publication.

Reviewer #2 (Remarks to the Author):

All my questions have been well addressed.